# Evaluation of ACCMIP ozone simulations and ozonesonde sampling biases using a satellite-based multi-constituent chemical reanalysis

Kazuyuki Miyazaki[1,2] and Kevin Bowman[2]

[1]Japan Agency for Marine-Earth Science and Technology, Yokohama 236-0001, Japan
[2]Jet Propulsion Laboratory-California Institute of Technology, Pasadena, CA, USA

*Correspondence to:* K. Miyazaki (kmiyazaki@jamstec.go.jp)

**Abstract.**

The Atmospheric Chemistry Climate Model Intercomparison Project (ACCMIP) ensemble ozone simulations for the present-day from the 2000 decade simulation results are evaluated by a state-of-the-art multi-constituent atmospheric chemical reanalysis that ingests multiple satellite data including the Tropospheric Emission Spectrometer (TES), the Microwave Limb Sounder (MLS), the Ozone Mapping Instrument (OMI), and the Measurements of Pollution in the Troposphere (MOPITT) for 2005–2009. Validation of the chemical reanalysis against global ozonesondes shows good agreement throughout the free troposphere and lower stratosphere for both seasonal and year-to-year variations, with an annual mean bias of less than 0.9 ppb in the middle and upper troposphere at the tropics and mid-latitudes. The reanalysis provides comprehensive spatio-temporal evaluation of chemistry-model performance that compliments direct ozonesonde comparisons, which are shown to suffer from significant sampling bias. The reanalysis reveals that the ACCMIP ensemble mean overestimates ozone in the northern extratropics by 6–11 ppb while underestimating by up to 18 ppb in the southern tropics over the Atlantic in the lower troposphere. Most models underestimate the spatial variability of the annual mean lower tropospheric concentrations in the extratropics of both hemispheres by up to 70 %. The ensemble mean also underestimates the seasonal amplitude by 25–70 % in the northern extratropics and overestimates the inter-hemispheric gradient by about 30 % in the lower and middle troposphere. A part of the discrepancies can be attributed to the five-year reanalysis data for the decadal model simulations. However, these differences are less evident with the current sonde network. To estimate ozonesonde sampling biases, we computed model bias separately for global coverage and the ozonesonde network. The ozonesonde sampling bias in the evaluated model bias for the seasonal mean concentration relative to global coverage is 40–50 % over the Western Pacific and East Indian Ocean and reaches 110 % over the equatorial Americas, and up to 80 % for the global tropics. In contrast, the ozonesonde sampling bias is typically smaller than 30 % for the Arctic regions in the lower and middle troposphere. These systematic biases have implications for ozone radiative forcing and the response of chemistry to climate that can be further quantified as the satellite observational record extends to multiple decades.

# 1 Introduction

Tropospheric ozone is one of the most important air pollutants and the third most important anthropogenic greenhouse gases in the atmosphere (Forster et al., 2007; HTAP, 2010; Myhre et al., 2013; Stevenson et al., 2013) while also playing a crucial role in the tropospheric oxidative capacity through production of hydroxyl radicals (OH) by photolysis in the presence of water vapor (Logan et al., 1981; Thompson, 1992). Global tropospheric ozone is formed from secondary photochemical production of ozone precursors including hydrocarbons or carbon monoxide (CO) in the presence of nitrogen oxides ($NO_x$) modulated by additional processes including in-situ chemical loss, deposition to the ground surface, and inflow from the stratosphere. These ozone precursors are largely controlled by anthropogenic and natural emission sources, e.g., transport, industry, lightning, biomass burning sources. Representation of tropospheric ozone in chemical transport models (CTMs) and chemistry climate models (CCMs) is also important in estimating its impact on the atmospheric radiative budget. A number of chemical transport models (CTMs) and chemistry climate models (CCMs) have been developed and used to study variations in atmospheric environment and its impacts on climate (e.g. Bowman et al., 2013; Shindell et al, 2006, 2013; Stevenson et al., 2006, 2013; Wild, 2007, Kawase et al., 2011; Young et al., 2013). However, current tropospheric ozone simulations still have large uncertainties because of the incomplete representation of model processes, as well as the large uncertainty in precursor emissions. These in turn increase uncertainty in CCM projections.

Climate model evaluation has primarily been achieved by comparisons with observed concentrations or related variables, which requires a precise description of their geographical, vertical, and temporal variations. Various measurements have been employed for evaluating simulated fields (e.g., Huijnen et al., 2010; Parrish et al., 2014; Stevenson et al., 2006, 2013; Young et al., 2013). However, information obtained from individual measurements is limited, and evaluation of global ozone fields with a suite of satellite measurements and in situ measurements is challenging because of limited vertical sensitivity profiles that differ among measurements, different overpass times, and mismatches in spatial and temporal coverage between the instruments. First, surface measurements have a spatial representativeness that is much smaller than that of global models over polluted areas. Second, ozone climatology data sets have been established based on ozonesonde measurements for use in model evaluation (Logan et al., 1999; Considine et al., 2008). Tilmes et al. (2012) generated an ozone climatology using ozonesonde measurements obtained between 1995 and 2011, which mostly consists of the same station data described by Logan (1999) and Thompson et al. (2003), but covering a longer time period. Using the compiled data of Tilmes et al. (2012), Young et al. (2013) conducted an intensive validation of tropospheric ozone from multiple model simulations in the Atmospheric Chemistry and Climate Model Inter-comparison Project (ACCMIP). However, the climatological data do not provide information on the temporal variability of the observed ozone. In addition, the current ozonesonde network does not cover the entire globe and is not homogeneously distributed between the hemispheres, ocean and land, and urban and rural areas, and its sampling interval is typically a week or longer. Model errors are also expected to vary greatly in time and space at various scales. Therefore, we consider that the spatial and temporal coverage of the ozonesonde network is insufficient to capture the temporally and spatially representative model bias. Third, satellite-retrieved measurements such as those from the Tropospheric Emission Spectrometer (TES) (Herman and Kulawik, 2013) and the Infrared Atmospheric Sounding Interferometer (IASI) (Clerbaux et al., 2009) have

great potential for evaluating global ozone fields (e.g., Aghedo et al., 2011). However, information obtained from currently available satellite measurements are still limited. Their vertical sensitivity is not strong enough to resolve detailed vertical structures in the troposphere as appeared in current global models, and they measure at only a particular overpass time, thus the diurnal variation information is missing. Meanwhile, the characteristics of each measurement, such as observational error, vary with observational condition, but their influence is rarely taken into consideration in model evaluations.

Data assimilation is a technique for combining different observational data sets with a model, with consideration of the characteristics of individual measurements (e.g., Kalnay, 2003; Lahoz and Schneider, 2014). Advanced data assimilation allows the propagation of observational information in time and space and from a limited number of observed species to a wide range of chemical components, and provides global fields that are physically and chemically consistent and in agreement with individual observations (Sandu and Chai, 2011; Bocquet et al., 2015). Various studies have demonstrated the capability of data assimilation techniques in the analysis of chemical species in the troposphere and stratosphere (e.g. Stajner and Wargan, 2004; Jackson, 2007; Parrington et al., 2009; Kiesewetter et al., 2010; Flemming et al., 2011; Coman et al., 2012; Inness et al., 2013; Emili et al., 2014; Miyazaki et al., 2012a, 2012b, 2013, 2014, 2015, 2016; van der A et al., 2015; Gaubert et al., 2016).

Reanalysis is a systematic approach to creating a long-term data assimilation product. Meteorological reanalyses have been established at operational centers for many years and are widely used in climate and meteorological research (e.g., Hartmann et al., 2013). Tropospheric chemical reanalysis, however, is relatively new. Inness et al. (2013) performed an eight-year reanalysis of tropospheric chemistry for 2003–2010 using a coupled system Integrated Forecast System coupled to the Model for OZone And Related chemical Tracers (IFS-MOZART) with observations sensitive primarily to the upper troposphere, and highlighted the importance of estimating surface emissions. This chemical reanalysis is recently updated by Flemming et al. (2017) using the integrated forecasting system with modules for atmospheric composition (C-IFS) with CB05 chemistry. Miyazaki et al. (2015) simultaneously estimated concentrations and emissions for an eight-year tropospheric chemistry reanalysis for 2005–2012 obtained from an assimilation of multi-constituent satellite measurements, which had greater lower tropospheric sensitivity, using an ensemble Kalman filter (EnKF). Chemical reanalysis using the EnKF has been used to provide comprehensive information on atmospheric composition variability and elucidate variations in precursor emissions and to evaluate bottom-up emission inventories (Miyazaki et al., 2014, 2015, 2016).

In this study, we explore the new potential of chemical reanalysis for evaluation of tropospheric ozone profiles in multi-model chemistry climate simulations from ACCMIP (Lamarque et al., 2013). Model errors in precursors can also be evaluated using the reanalysis product, and this could help identify error sources in tropospheric ozone simulations. However, because no other study has shown the potential of reanalysis ozone for model evaluation, this study focuses on tropospheric ozone only. ACCMIP models have been used to calculate historic and future radiative and chemically important species and their coupling with the broader climate system (Bowman et al., 2013; Lee et al., 2013; Naik et al., 2013; Stevenson et al., 2013; Shindell et al., 2013; Voulgarakis et al., 2013; Young et al., 2013). We characterize ACCMIP models in simulating global distributions and the seasonal variation of ozone from the lower troposphere to the lower stratosphere. We further discuss the limitation of the current ozonesonde network for evaluating temporally and spatially representative model errors. To the best of our knowledge, this is the first study to apply chemical reanalysis to the evaluation of global chemistry-climate mod-

els and consequently offers a similar potential as meteorological reanalysis for evaluation of climate models (Ana4MIPS, https://esgf.nccs.nasa.gov/projects/ana4mips/Background).

## 2 Methodology

### 2.1 Chemical data assimilation system

5 The data assimilation system is constructed based on a global CTM MIROC-Chem (Watanabe et al. 2011) and an EnKF described in Miyazaki et al. (2016), which can be consulted for more detailed information. We use the two-hourly global chemical reanalysis data for the period 2005–2009 when tropospheric ozone fields are strongly constrained by TES tropospheric ozone measurements. The availability of TES measurements is strongly reduced after 2010, which led to a degradation of the reanalysis performance, as demonstrated by Miyazaki et al. (2015).

10 A major update from the system used in Miyazaki et al. (2015) to the system used in this study is the replacement of forecast model from CHASER (Sudo et al., 2002) to MIROC-Chem (Watanabe et al., 2011), which caused substantial changes in the a priori field and thus the data assimilation results of various species. Microwave Limb Sounder (MLS) retrievals have been updated from v3.3 in Miyazaki et al. (2015) to v4.2 in this study. In addition, we attempt to optimize the surface $NO_x$ emission diurnal variability using data assimilation of multiple $NO_2$ satellite retrievals obtained at different overpass times in the updated 15 system (Miyazaki et al., 2016).

#### 2.1.1 Forecast model

The forecast model, MIROC-Chem (Watanabe et al., 2011), considers detailed photochemistry in the troposphere and stratosphere by simulating tracer transport, wet and dry deposition, and emissions, and calculates the concentrations of 92 chemical species and 262 chemical reactions (58 photolytic, 183 kinetic, and 21 heterogeneous reactions). Its tropospheric chemistry 20 considers the fundamental chemical cycle of $O_x$-NOx-$HO_x$-$CH_4$-CO along with oxidation of non-methane volatile organic compounds (NMVOCs) to properly represent ozone chemistry in the troposphere. Its stratospheric chemistry simulates chlorine and bromine containing compounds, CFCs, HFCs, OCS, $N_2O$, and the formation of polar stratospheric clouds (PSCs) and associated heterogeneous reactions on their surfaces. The radiative transfer scheme considers absorption within 37 bands, scattering by gases, aerosols, and clouds, and the effect of surface albedo. Detailed radiation calculations are used for photolysis 25 calculation. Methane concentrations were scaled on the basis of present-day values with reference to the surface concentration. MIROC-Chem has a T42 horizontal resolution ($2.8°$) with 32 vertical levels from the surface to 4.4 hPa. The horizontal model resolution is comparable to the resolution of ACCMIP models (ranging from $1.24°$ to $5°$). It is coupled to the atmospheric general circulation model MIROC-AGCM version 4 (Watanabe et al., 2011). The simulated meteorological fields were nudged toward the six-hourly ERA-Interim (Dee et al., 2011) to reproduce past meteorological fields.

30 The a priori values for surface emissions of $NO_x$ and CO were obtained from bottom-up emission inventories. Anthropogenic $NO_x$ and CO emissions were obtained from the Emission Database for Global Atmospheric Research (EDGAR)

version 4.2 (EC-JRC, 2011). Emissions from biomass burning were based on the monthly Global Fire Emissions Database (GFED) version 3.1 (van der Werf et al., 2010). Emissions from soils were based on monthly mean Global Emissions Inventory Activity (GEIA) (Graedel et al., 1993). Lightning $NO_x$ ($LNO_x$) sources in MIROC-Chem were calculated based on the relationship between lightning activity and cloud top height (Price and Rind, 1992) and using the convection scheme of MIROC-AGCM developed based on the scheme presented by Arakawa and Schubert (1974). For black carbon (BC) and organic carbon (OC) and other precursor gases, surface and aircraft emissions are specified from the emission scenarios for Greenhouse Gas and Air Pollution Interactions and Synergies (GAINS) model developed by International Institute for Applied System Analysis (IIASA) (Klimont et al., 2009; Akimoto et al., 2015).

### 2.1.2  Data assimilation method

Data assimilation used here is based upon on an EnKF approach (Hunt et al., 2007). The EnKF uses an ensemble forecast to estimate the background error covariance matrix and generates an analysis ensemble mean and covariance that satisfy the Kalman filter equations for linear models. In the forecast step, a background ensemble, $x_i^b (i = 1, ..., k)$, is obtained from the evolution of an ensemble model forecast, where $x$ represents the model variable, $b$ is the background state, and $k$ is the ensemble size (i.e., 32 in this study). The ensemble perturbations were introduced to all the state vector variables as described below. The background ensemble is then converted into the observation space, $y_i^b = H(x_i^b)$, using the observation operator $H$ which is composed of a spatial interpolation operator and an operator that converts the model fields into retrieval space, which can be derived from an a priori profile and an averaging kernel of individual measurements (e.g., Eskes and Boersam, 2003; Jones et al, 2003). Using the covariance matrices of observation and background error as estimated from ensemble model forecasts, the data assimilation determines the relative weights given to the observation and the background, and then transforms a background ensemble into an analysis ensemble, $x_i^a (i = 1, ..., k)$. The new background error covariance is obtained from an ensemble forecast with the updated analysis ensemble.

In the data assimilation analysis, a covariance localization is applied to neglect the covariance among unrelated or weakly related variables, which has the effect of removing the influence of spurious correlations resulting from the limited ensemble size. The localization is also applied to avoid the influence of remote observations that may cause sampling errors. The state vector includes several emission sources (surface emissions of $NO_x$ and CO, and $LNO_x$ sources) as well as the concentrations of 35 chemical species. The emission estimation is based on a state augmentation technique, in which the background error correlations determines the relationship between the concentrations and emissions of related species for each grid point. Because of the simultaneous assimilation of multiple-species data and because of the simultaneous optimization of the concentrations and emission fields, the global distribution of various species, including OH, is modified considerably in our system. Miyazaki et al. (2015) demonstrated that the Northern/Southern Hemisphere OH ratio became closer to an observational estimate of Patra et al. (2014) due to the multiple-species assimilation. This propagates the observational information between various species and modulates the chemical lifetimes of many species (Miyazaki et al., 2012b; 2015; 2016).

### 2.1.3 Assimilated measurements

Assimilated observations were obtained from multiple satellite measurements (Table 1). Tropospheric $NO_2$ column retrievals used are the version-2 Dutch Ozone Monitoring Instrument (OMI) $NO_2$ (DOMINO) data product (Boersma et al., 2011) and version 2.3 TM4NO2A data products for Scanning Imaging Absorption Spectrometer for Atmospheric Cartography (SCIA-MACHY) and Global Ozone Monitoring Experiment-2 (GOME-2) (Boersma et al., 2004) obtained through the TEMIS website (www.temis.nl). The TES ozone data and observation operators used are version 5 level 2 nadir data obtained from the global survey mode (Bowman et al, 2006; Herman and Kulawik, 2013). This data set consists of 16 daily orbits with a spatial resolution of 5–8 km along the orbit track. The MLS data used are the version 4.2 ozone and $HNO_3$ level 2 products (Livesey et al., 2011). We used data for pressures of less than 215 hPa for ozone and 150 hPa for $HNO_3$. The Measurement of Pollution in the Troposphere (MOPITT) CO data used are version 6 level 2 TIR products (Deeter et al., 2013).

### 2.2 ACCMIP models

The Atmospheric Chemistry Climate Model Intercomparison Project (ACCMIP) focuses on chemistry-climate interactions needed to compute the proper climate forcing for Climate Model Intercomparison Project (CMIP5) climate simulations (Taylor et al., 2012) as well as the impact of climate change on chemical species. The ACCMIP consists of a series of time slice experiments for the long-term changes in atmospheric composition between 1850 and 2100, as described by Lamarque et al. (2013). The experimental design was based on decadal time-slice experiments driven by decadal mean sea surface temperatures (SST). This study uses the 2000 decade simulation results from 15 models (1. CESM-CAM, 2. CICERO-OsloCTM2, 3. CMAM, 4. EMAC, 5. GEOSCCM, 6. GFDL-AM3, 7. GISS-E2-R, 8. GISS-E2-TOMAS, 9. HadGEM2, 10. LMDzOR-INCA, 11. MIROC-CHEM, 12. MOCAGE, 13. NCAR-CAM3.5, 14. STOC-HadAM3, 15. UM-CAM). The number of years that the ACCMIP models simulated for the 2000 decadal simulation mostly varied between 4 and 12 years for each model. Each model simulation was averaged over the simulated years.

Meteorological fields were obtained from analyses in CICERO-OsloCTM2 and from climate model fields in MOCAGE. UM-CAM and STOC-HadAM3 simulated meteorological and chemical fields, but chemistry did not affect climate. In all other models, simulated chemical fields were used in the radiation calculations and hence provide a forcing effect on the general circulation of the atmosphere. Lamarque et al. (2013) indicated that most models overestimate global annual precipitation and have a cold bias in the lower troposphere.

Different models vary greatly in complexity. The calculated chemical species vary from 16 to 120 species. Photolysis rates are computed with offline or online methods, depending on the model. Many models include a full representation of stratospheric ozone chemistry and the heterogeneous chemistry of polar stratospheric clouds, but several models specify stratospheric ozone. Methane concentration is prescribed for the surface or over the whole atmosphere in many models. Ozone precursor emissions from anthropogenic and biomass burning sources were taken from those compiled by Lamarque et al. (2010). Natural emission sources such as isoprene emissions, and lightning and soil $NO_x$ sources were not specified and were accounted for differently between models. There is a large range in soil $NO_x$ emissions from 2.7 to 9.3 $TgNyr^{-1}$ and in $LNO_x$ sources

from 1.2 to 9.7 TgNyr$^{-1}$ for the 2000 conditions. The range of natural emissions is a significant source of model-to-model ozone differences (Young et al. 2013). A complete description of the models along with the experiment design can be found in Lamarque et al. (2013).

Both the ACCMIP models and chemical reanalysis are interpolated to at $2° \times 2.5°$ spatial resolution and 67 levels, following Bowman et al. (2013), and then compared each other. Spatial correlations are computed with consideration of weighting for the latitude.

## 2.3  Ozonesonde data

Ozonesonde observations were taken from the World Ozone and Ultraviolet Radiation Data Center (WOUDC) database (available at http://www.woudc.org). All available data from the WOUDC database are used for the evaluation of reanalysis data (Section 3), as listed in Table 2. For the evaluation of ACCMIP models and ozonesonde sampling biases (Section 4 and 5), we use the ozonesonde sampling based on the compilation by Tilmes et al. (2012), which is shown in bold in Table 2. Because there is no observation after 2003 in Scoresbysund, this location has been removed from the compilation in this study. The accuracy of the ozonesonde measurement is about $\pm 5$ % in the troposphere (Smit and Kley, 1998).

To compare ozonesonde measurements with the data assimilation and ACCMIP models, all ozonesonde profiles have been interpolated to a common vertical pressure grid, with a bin of 25 hPa. The two-hourly reanalysis and forecast model (i.e., control run) fields were linearly interpolated to the time and location of each measurement, with a bin of 25 hPa, and then compared with the measurements. For the ACCMIP models, the monthly model outputs were compared with the measurements at the location of each measurement. The averaged profile is computed globally and for four latitudinal bands, SH extratropics (90–30° S), SH tropics (30° S–Equator), NH tropics (Equator–30° N), and NH extratropics (30–90° N).

## 2.4  Ozonesonde sampling bias estimation

The current ozonesonde network does not cover the entire globe and is not homogeneously distributed between the hemispheres, ocean and land, and urban and rural areas. Also, the sampling interval of ozonesonde observations is typically a week or longer, which does not reflect the influence of diurnal and day-to-day variations. Model errors are also expected to vary greatly in time and space at various scales. Therefore, the implications of model differences at ozonesonde locations to regional and seasonal processes is uncertain. Thus, we evaluate how changes in evaluated model performance could be obtained by using the complete sampling chemical reanalysis fields instead of the existing ozonesonde network on simulated regional ozone fields.

Sampling bias is an error in a computed quantity that arises due to unrepresentative (i.e., insufficient or inhomogeneous) sampling, which induces spurious features in the average estimates (e.g., Aghedo et al., 2011; Foelsche et al 2011; Toohey et al., 2013; Sofieva et al., 2014) and long-term trends (Lin et al., 2016). Sampling bias may occur when the atmospheric state within the time-space domain over which the average is calculated is not uniformly sampled. In regions where variability is dominated by short-term variations, limited sampling may lead to a random sampling error. The primary technique for sampling bias estimation is to subsample model or reanalysis fields based on the sampling patterns of the measurements and

then to quantify differences between the mean fields based on the measurement sampling and those derived from the complete fields. Sampling bias cannot be negligible, even for satellite measurements (Aghedo et al. 2011; Toohey et al., 2013; Sofieva et al., 2014).

To estimate sampling biases of the ozonesonde network in the ACCMIP model evaluation, two evaluation results of mean model bias are compared using the chemical reanalysis. The first evaluation was conducted based on the complete sampling; the second evaluation used the ozonesonde sampling (in both space and time) that is based on the compilation by Tilmes et al. (2012). By using the two-hourly reanalysis fields, we can address possible biases due to the limited model sampling (i.e., monthly ACCMIP model outputs were used). Note that the relatively coarse horizontal resolution of the reanalysis may lead to an underestimation of the sampling bias in the model evaluation, because the variability of a sampled field depends on the resolution of the measurement. Tilmes et al. (2012) stated that regional aggregates of individual ozonesonde measurements with similar characteristics are more representative for larger regions; however, this may not mean that evaluation results using the compiled data generate model errors that are representative of actual monthly mean for a surrounding area.

## 3   Consistency between chemical reanalysis and ozonesonde observations

Miyazaki et al. (2015) validated an older version of the reanalysis (http://www.jamstec.go.jp/res/ress/kmiyazaki/reanalysis/) and showed good agreement with independent observations such as ozonesonde and aircraft measurements on regional and global scales and for both seasonal and year-to-year variations from the lower troposphere to the lower stratosphere for the 2005-2012 period. The mean bias against the ozonesonde measurements in the older dataset is -3.9 ppb at the NH high-latitudes (55–90° N), -0.9 ppb at the NH mid-latitudes (15–55° N), 2.8 ppb in the tropics (15° S–15° N), -1.0 ppb at the SH mid-latitudes (55–15° S), -1.7 ppb at the SH high-latitudes (90–55° S) between 850 and 500 hPa (Miyazaki et al., 2015). Since the updated reanalysis ozone fields used in this study have not yet been validated in any publication, we first present the evaluation results of the chemical reanalysis using global ozonesonde observations for 2005–2009.

Figs 1 and 2 compare the reanalysis and the global ozonesonde observations, and the comparison result is summarized in Table 3. In order to confirm improvements in the reanalysis, results from a model simulation without any chemical data assimilation (i.e., a control run) is also shown. The control run shows systematic biases, such as positive biases in the upper troposphere and lower stratosphere (UTLS) throughout the globe and negative biases in the lower and middle troposphere in the extratropics of both hemispheres. The positive bias in the UTLS is larger in the Southern Hemisphere (SH) than in the Northern Hemisphere (NH). The a priori systematic bias in this study is larger than that in our previous study (Miyazaki et al., 2015) in the UTLS, because of different model settings, such as the upper boundary conditions of $NO_y$, $Cl_y$, and $Br_y$. However, the reanalysis fields were less sensitive to the a priori profiles in the UTLS than in the lower and middle troposphere because of strong constraints by MLS measurements and long chemical lifetime of ozone in the UTLS.

The reanalysis shows improved agreements with the ozonesonde observations over the globe for most cases. The data assimilation removed most of the positive bias in the UTLS throughout the year and reduced the negative bias in the lower and middle troposphere in the extratropics. In the NH extratropics in the lower and middle troposphere, the data assimilation reduced the

annual mean negative bias of the forecast model by 55 %, which is attributed to the reduced bias in boreal spring–summer. The mean bias in the new reanalysis dataset is smaller than that in the older reanalysis dataset (Miyazaki et al., 2015) for most cases (e.g., from -3.9 to -2.9 ppb at the NH high-latitudes (55–90° N), -0.9 to -0.1 ppb at the NH mid-latitudes (15–55° N), -1.0 to -0.1 ppb at the SH mid-latitudes (55–15° S) between 850 and 500 hPa). The mean bias in the new dataset is less than 0.9 ppb at the tropics and mid-latitudes between 500 and 200 hPa (not shown). The simultaneous optimization of concentrations and emissions played important roles in improving the lower tropospheric ozone analysis, associated with the pronounced ozone production caused by $NO_x$ increases, as demonstrated by Miyazaki et al. (2015). This advantage increases the ability of the chemical reanalysis to evaluate the simulated tropospheric ozone profiles, including the lower tropospheric ozone concentrations. Root-Mean-Square-Errors (RMSEs) are also reduced above the middle troposphere, although the reduction rate is relatively small compared to the bias, probably due to representativeness errors between the ozonesonde measurements and data assimilation analysis. The tropospheric concentrations show distinct seasonal and year-to-year variations, for which the temporal correlation based on the monthly and regional mean concentrations is increased by the data assimilation globally, except at high latitudes in the lower troposphere (Table 3). The reanalysis can be extended to a longer-term validation that will provide more information on seasonality and year-to-year variability.

## 4 Evaluation of ACCMIP models

### 4.1 Global distribution

We use the global chemical reanalysis to evaluate the global ozone profiles in ACCMIP simulations. Fig. 3 compares the global distribution of the annual mean ozone concentration between the five-year mean reanalysis and the ensemble mean of the AC-CMIP models. The average over the multiple models can be expected to improve the robustness of the model simulation results, because some parts of the model errors may cancel each other out. As summarized in Table 4, the global spatial distributions are similar between the five-year mean reanalysis field and the ensemble mean when estimated at 2°×2.5° spatial resolution, with a spatial correlation (r) greater than 0.94 from the lower troposphere to the lower stratosphere, except for the NH extratropical middle troposphere (r=0.57). The reanalysis and multi-model mean commonly reveal distinct inter-hemispheric differences, associated with a stronger downwelling across the tropopause and stronger emission sources of ozone precursors in the NH. The wave-1 pattern in the zonal ozone distribution in the tropics, with a minimum over the Pacific Ocean and maximum over the Atlantic (Thompson et al., 2003; Bowman et al, 2009; Ziemke et al., 2011), can also be commonly found in the reanalysis and the multi-model mean and was also suggested by Young et al. (2013).

Large errors between the reanalysis and the multi-model mean in the troposphere are found in the NH extratropics and SH tropics (right panel in Fig. 3). The multi-model mean overestimates the zonal and annual mean concentrations by 6–11 ppb at 800 hPa and by 2–9 ppb at 500 hPa in the NH extratropics. The overestimation is larger over the oceans than over land at the NH mid-latitudes at 800 hPa. Both the mean RMSE and bias are larger at 800 hPa than at 500 hPa in the NH extratropics, whereas they are larger at 500 hPa in the NH tropics (Table 4). In the SH tropics, the multi-model mean underestimates the concentration over the eastern Pacific by up to 9 ppb, over the Atlantic by up to 18 ppb, and over the Indian Ocean by up to 8

ppb at 500 hPa. These negative biases are larger in the middle troposphere than in the lower troposphere for most places and also for the zonal means in the SH tropics (-15 % in the middle troposphere and -10 % in the lower troposphere) (Table 4). Young et al. (2013) consistently revealed the positive bias in the NH and negative bias in the SH using OMI/MLS tropospheric ozone column measurements. At 200 hPa, the multi-model mean underestimates the zonal mean concentration by 20–30 ppb at high latitudes in both hemispheres, with a larger error in the SH than in the NH (Table 4).

Fig. 4 shows the Taylor diagram of the ACCMIP models against the reanalysis for three latitudinal bands for three levels. The relevant statistics at 500 hPa are summarized in Table 5, for which the tropics is separated into two hemispheres. In the NH extratropics at 800 hPa, most models reproduced the spatial distribution (r = 0.8–0.95), while underestimating the spatial standard deviation (SD) by up to 50 %. Three exceptional models (1, 7, 8) show relatively poor agreements (r = 0.45–0.6 and SD underestimations by 50–60 %). At 500 hPa, there is a large diversity in the agreement. Only a few models (2, 4, 9, 11) show close agreement with the reanalysis (r > 0.8, SD error < 20 %). Notably, two models (12, 15) reveal too large spatial variabilities (SD error > 80 %), and five models (1, 6, 7, 8, 12) reveal small spatial correlation (r < 0.15). The regional mean bias is largely positive (> 10 ppb) in several models (7, 8, 12) (Table 5). In the NH extratropics in the lower and middle troposphere, ozone distributions are modified by various processes, including vertical transport by convection and along conveyor belts, inflow from the stratosphere, long-range transports, and photochemical production (e.g, Lelieveld and Dentine, 2000; Oltmants et al., 2006; Sudo and Akimoto, 2007; Jonson et al., 2010). The evaluation results indicate that these processes occur differently among models. At 200 hPa, all the models simulate well the spatial distribution (r > 0.95), whereas the spatial variability differs between the models (SD error ranges from -50 % to +30 %). There is relatively large variation in the stratospheric concentration, which results in the diversity in the UTLS, as also discussed by Young et al. (2013).

In the tropics, the spatial correlation is greater than 0.8 at all levels for most models (except for 12, 15), as they capture the wave-1 structure. When dividing the tropics into two hemispheres (Table 5), only a few models (4, 12) reveal low spatial correlation (r < 0.8) for the SH tropics (30°S–EQ) at 500 hPa. The spatial correlation in the tropics is lower at 500 hPa than at 800 hPa for most models. The SD error is less than 40 % for all the models at 800 and 500 hPa, while mostly overestimating the spatial variability at 800 hPa by up to 30 %. The mean bias is negative for most models at 500 hPa in the tropics in both hemispheres, with larger negative biases in the SH tropics (Table 5). Young et al (2013) noted that correlations between the biases for the NH and SH tropical tropospheric columns are strong. Similarly, our analysis using the reanalysis reveal a high correlation (0.91) between the NH and SH tropical biases at 500 hPa, suggests that similar processes are producing the model biases in the tropical middle troposphere between the hemispheres. For instance, biomass burning emissions are handled differently across the models, which may lead to differences in ozone simulations in the tropics (Anderson et al., 2016). At 200 hPa in the tropics, the SD error differs among models, which could primarily be associated with the different representations of convective transports and ozone production by $LNO_x$ sources (e.g., Lelieveld and Crutzen, 2007; Wu et al., 2007).

In the SH extratropics at 800 hPa, most models reproduce the spatial distribution (r > 0.9), while underestimating the SD by 15–70 %, except for model 15. The model performance is similar between 800 hPa and 500 hPa, with a smaller SD error at 500 hPa for most models. These high spatial correlations may be related to a lack of local precursor emissions in the SH. At 500 hPa, a majority of the models underestimate the mean concentration (Table 5), with large negative biases (< -8 ppb) in several

models (1, 2, 12, 14). At 200 hPa, the SD error varies from -80 % to + 65 %. The large diversity at 200 hPa may be related to the different representation of the tropopause and stratosphere–troposphere exchange (STE) among models.

## 4.2  Seasonal variation

Fig. 5 compares the seasonal variation of zonal mean ozone concentration between the ACCMIP models, the reanalysis, and
ozonesonde observations. The comparison between the reanalysis concentrations sampled at ozonesonde sites/time (black dashed line) and the ozonesonde observations (blue solid line) shows that the reanalysis is in close agreement with the ozonesonde observations over the globe, as described in Sec. 3. However, in the NH extratropics at 800 hPa, the reanalysis concentration is too low from boreal spring to summer by up to 4 ppb, which leads to an underestimation of the seasonal amplitude (as estimated from the difference between maximum and minimum monthly mean concentrations). In the NH tropics
at 500 hPa, the reanalysis overestimates the concentration except in April. In the SH tropics at 500 and 800 hPa, the reanalysis slightly overestimates the concentrations throughout the year by up to 5 ppb. In the SH extratropics at 800 hPa, the reanalysis concentration is too low by up to 5 ppb from austral autumn to winter. The reanalysis concentration and seasonal variation differs largely between the complete sampling (black bold line, where the concentrations were averaged over all grid points) and the ozonesonde sampling (black dashed line) for the globe. The impact of using the reanalysis instead of the ozonesonde
network in characterizing the ozone seasonal variation is discussed in Section 5.

The global ozone concentrations averaged over all grid points with area weights are compared between the ACCMIP models and the reanalysis (black solid line vs. red solid line for the multi-model mean and thin colored lines for individual models). There is considerable interannual variability in both the reanalysis and the ACCMIP models. We confirmed that the ACCMIP ensemble mean is mostly within the standard deviation (i.e., year-to-year variation) of the reanalysis (not shown). In the NH
extratropics, the multi-model mean overestimates the monthly mean concentrations by 6–9 ppb at 800 hPa and by 3–6.5 ppb at 500 hPa. The multi-model mean reproduces the seasonal variation, whereas there is large diversity among the models. The increase from winter to spring differs among models at 500 hPa, which is probably associated with different representations of downwelling from the stratosphere. Fig. 6 compares the seasonal amplitude. Most models overestimate the seasonal amplitude in the NH lower and middle troposphere, with a mean overestimation of 50–70 % at 800 hPa and 25–40 % at 500 hPa at
NH high latitudes. At 200 hPa, the multi-model annual mean concentration is in good agreement with that of the reanalysis, whereas the seasonal amplitude is underestimated by most models at NH high latitudes, with a mean underestimation of 15–25 %.

In the NH tropics at 500 hPa, the multi-model mean underestimates the concentration by 1–4 ppb throughout the year, which can be attributed to the anomalously low concentrations in several models. There is a large diversity among the models in this
region. In the SH subtropics, the multi-model mean is lower by up to 5 ppb at 800 hPa and by up to 11 ppb at 500 hPa, with the largest errors occurring in austral spring. A majority of models overestimate the seasonal amplitude in the NH subtropics at 800 hPa (by about 10–40 %), whereas they mostly underestimate the amplitude in the SH tropics at 800 and 500 hPa. In the tropical upper troposphere in both hemispheres, a few models reveal anomalously high or low concentrations. Both the ozonesondes

and reanalysis reveal a sharp increase in ozone between March and April in the NH subtropics, which is not captured in the multi-model mean, as suggested by Young et al. (2013).

In the SH extratropics, the multi-model mean and the reanalysis are in good agreement at 800 hPa, whereas it largely underestimates the peak concentration in austral winter–spring at 500 hPa (by up to 7 ppb) and 200 hPa (by up to 35 ppb). The large diversity among the models and the large underestimation in the multi-model mean at 500 hPa in spring could be attributed to the differing influence of stratospheric air. The seasonal amplitude is overestimated at 800 and 200 hPa by most models at SH high-latitudes.

## 4.3 Inter-hemispheric gradient

Fig. 7 compares the inter-hemispheric gradient (NH/SH ratio) of the annual mean ozone concentration. We calculated the gradient of area-weighted ozone concentrations across the equator; however, recognize a more careful definition of the boundary between two hemispheres would be required to isolate air masses originated from each hemisphere (e.g., Hamilton et al., 2008). For the estimation of the gradient using the ozonesonde observations, we made a gridded dataset from the ozonesonde observations based on the completion by Tilmes et al (2012) at $2° \times 2.5°$ spatial resolution, and then calculated area-weighted hemispheric mean concentrations using the gridded dataset. The gradient is similar between the ozonesonde observations (blue solid line) and the reanalysis concentration from the ozonesonde sampling (black dashed line) throughout the troposphere. In these estimates, the NH mean concentration is higher than the SH mean by 60–70 % in the lower troposphere, by 30–40 % in the middle troposphere, and by 55–60 % around 200 hPa. Near the surface, the reanalysis slightly overestimates the NH/SH ratio, mainly because of overestimated concentrations at the NH mid-latitudes.

By taking a complete sampling in the reanalysis (i.e., averaging over all model grid points for each hemisphere) (black solid line), the NH/SH ratio becomes smaller by about 25–30 %, 7–10 %, and 15–25 % in the lower troposphere, the middle troposphere, and around 200 hPa, respectively, compared to the average at the ozonesonde sampling sites (black dashed line). The difference is a consequence of ozonesonde stations located near large cities at NH mid-latitudes, and therefore tend to observe higher ozone concentration than the hemispheric average. At around 200 hPa, the difference could also be attributed to the presence of atmospheric stationary waves and Asian monsoon circulation in the NH, which result in substantial spatial ozone variations in the UTLS (e.g., Wirth, 1993; Park et al., 2008) (c.f., Fig. 3). The annual mean NH/SH ratio based on the global reanalysis field estimated at the surface, 800 hPa, 500 hPa, and 200 hPa are 1.36, 1.42, 1.30, and 1.35, respectively.

Most models overestimate the NH/SH ratio compared with the reanalysis, with a mean overestimation (black solid line vs. red solid line) of 34 % at the surface and 22–30 % in the free troposphere, attributing to both too-high concentrations in the NH extratropics and too-low concentrations in the SH subtropics in most models (c.f., Figs. 3 and 5). The multi-model mean reveals annual mean NH/SH ratios of 1.71, 1.73, 1.54, and 1.49 at the surface, 800 hPa, 500 hPa, and 200 hPa, respectively. The large systematic error in the NH/SH ratio suggests that, for instance, the inter-hemispheric distribution of radiative heating due to tropospheric ozone in chemistry–climate simulations are largely uncertain in most models, and such comprehensive information for different altitudes in the troposphere cannot be obtained using any individual measurements, as is further discussed in Section 6.3.

## 5   Impact of sampling on model evaluation

As presented in the previous section, the chemical reanalysis provides comprehensive information on global ozone distributions for the entire troposphere which is useful for validating global model performance. It was also demonstrated that the inter-hemispheric gradient of ozone measured with the ozonesonde and complete sampling method produced different results, and the model-reanalysis difference strongly depended on the choice of the sampling method. As these networks have been the primary basis for CCM evaluation (e.g., Stevenson et al., 2006; Huijnen et al., 2010; Young et al., 2013), the implications of this sampling bias need to be quantified. This section evaluates how changes in evaluated model performance could be obtained by using the complete sampling chemical reanalysis fields instead of the existing ozonesonde network on simulated regional ozone fields.

The model evaluation results are shown for the 11 regions illustrated in Fig. 8 and summarized in Table 6. Japan was excluded from the evaluation because data from only one station was available for the reanalysis period. The 11 areas surrounding the ozonesonde stations were considered for complete atmospheric sampling (rectangles in Fig. 8), for which small margins were considered around the stations to prevent overestimation of the ozonesonde network limitation. It was confirmed that the discrepancy between the two evaluations generally increases with the size of the area. In contrast, for the SH mid- and high latitudes, the defined areas cover the entire range of longitudes, because of generally less variabilities in the SH than in the NH. Four latitude bands (90–30° S, 30° S–Equator, Equator–30° N, 30–90° N) were also considered in the sampling bias evaluation.

The reality of the reanalysis fields is important for reasonable estimates of the true sampling bias of the real atmosphere. As discussed in Section 3, there is good agreement in the evaluated model performance using the reanalysis and the ozonesonde measurements at the ozonesonde sampling, except for the lower troposphere. This result supports the use of the reanalysis data at the ozonesonde locations. The performance of the ACCMIP model as compared with the ozonesonde measurements is mostly consistent with that shown by Young et al. (2013), although the ozonesonde data periods differ – 1997-2011 was used by Young et al (2013) and 2005-2009 was used in this study.

Table 7 demonstrates the regional and seasonal mean differences of the reanalysis concentrations between the complete sampling and the ozonesonde sampling. The ozonesonde sampling results have higher concentrations (by about 3 %) in the two NH polar regions for most cases, whereas the difference is smaller in NH polar west than in NH polar east. Among the NH mid-latitude regions, a large difference (about 14 %) exists between the two cases over the eastern United States in June–August (JJA), where the comparison using monthly reanalysis fields sampled at the ozonesonde locations (brackets in Table 7) suggests that the sampling bias is dominated by temporal variations. The tropical and subtropical regions exhibit large sampling biases, 4–12.3 % over the NH subtropics, -3.2–5.0 % over the Western Pacific and East Indian Ocean, 0–7.8 % over the equatorial Americas, and -3.8–7.5 % over the Atlantic Ocean and Africa. In most of the tropical and subtropics regions, both the spatial and temporal sampling biases are important, because of large spatial and temporal variability of ozone and the sparse observation network. For the global tropics, the sampling bias reaches 13 % in the NH (Eq–30° N) and 8 % in the SH (30° S–Eq). Thus, the ozonesonde network has a major limitation when it comes to capturing ozone concentrations that are representative of seasonal and regional means for the entire tropical region. The sampling bias may not be negligible even in the

SH (0.3–3.9 % in the SH mid-latitudes and 0.8–4.2 % in the SH high latitudes), and it is large (up to 13 %) when estimations are done for a large area (90–30° S). The large sampling bias in 90–30° S is primarily attributed to spatial variability. The impact of the sampling bias on the model evaluation is discussed in the following section.

## 5.1 Mean error and its distribution

The model evaluation results differ greatly for many regions between the complete sampling and the ozonesonde sampling, as shown by Fig. 9 and summarized in Table 8. The sampling bias is evaluated using the median of the multiple models to provide robust estimates of the model performance. For the NH Polar Regions, Tilmes et al. (2012) stated that separating the regions into eastern and western sectors reduces the variability in ozone within each region because long-range transports of pollution from low and mid-latitudes into high latitudes shows longitudinal variations in the NH (e.g., Stohl, 2006). Compar-
isons further suggest that, except for the UTLS in winter (December–February (DJF)), the evaluated model performance using the ozonesonde measurements are representative of the surrounding regional and seasonal mean model performance. For the two NH polar regions at 200 hPa in DJF, the validation based on the ozonesonde sampling reveals a large negative sampling bias in the model bias as compared with regional and monthly means. Large negative model biases against the ozonesonde observations have been reported by Young et al. (2013) for 250 hPa (by about -13 % for the NH polar west and -18 % for
the NH polar east for the annual mean concentration), whereas results from this study suggest that these errors based on the ozonesonde sampling (by -14 % for the NH polar west and -18 % for the NH polar east in DJF in our estimates) are larger than those from regional and seasonally representative model bias (by -3 % and +5 %, respectively). At 500 hPa, the ozonesonde network reveals a negative sampling bias for the NH polar east in DJF. Thus, the positive bias reported in Young et al. (2013) for the NH polar east at 500 hPa may be lower than regional and seasonally representative model biases. Our analysis using
monthly reanalysis fields sampled at the ozonesonde locations (brackets in Table 8) suggests a greater impact of the spatial sampling bias than the temporal sampling bias for the NH polar east in DJF. The large discrepancy between the two estimates in the UTLS model performance can be attributed to the large variability of ozone distribution and associated model errors on a regional and seasonal scale.

For Canada, large differences (>30 %) exist in the two evaluations in the lower troposphere and for the UTLS in DJF and
for the middle troposphere in March–May (MAM). The ozonesonde measurements reveal a large negative sampling bias in the model evaluation in DJF at 200 hPa (-4 % in the complete sampling and -25 % in the ozonesonde sampling), while they reveal a negative sampling bias (by about 50 %) at 500 hPa in MAM. At 500 hPa over Canada, the relative importance of the spatial and temporal sampling biases varies with season: the spatial (temporal) sampling bias is dominant in DJF (JJA), whereas both of them are important in MAM. Similar differences between the two evaluations are found for Western Europe
at 500 hPa and at 200 hPa in DJF. These results suggest that, for instance, the positive bias for Western Europe estimated by Young et al (2013) may be lower than regional and seasonally representative model bias, even for such a small area. The smaller discrepancy between the two estimates for Western Europe as compared for Canada for most cases could be associated with the better coverage of the ozonesonde measurements for Western Europe. Even for the small area of the eastern United States, the two validations differ largely in the UTLS (e.g., -9 % in the ozonesonde sampling and +6 % in the complete sampling at

200 hPa in MAM) and at 500 hPa in MAM, JJA, and September–November (SON). In the NH subtropics, the two evaluations disagree largely in the middle and upper troposphere in JJA and SON.

The tropical stations were separated into the three sub-regions: Western Pacific and East Indian Ocean, equatorial America, and the Atlantic Ocean and Africa. These regions reflect the different dominant tropical processes including biomass burning and lightning over the Atlantic and Africa. The large variability of tropical ozone and its associated model error, together with the sparse ozonesonde network in these regions, results in large discrepancies between the two evaluations in the tropical regions. At 500 hPa, the ozonesonde measurements reveal a large (by 40–50 %) negative sampling bias in MAM and a positive sampling bias in DJF over the Western Pacific and East Indian Ocean, whereas it shows a large negative sampling bias (by 110 %) in MAM over the equatorial Americas. Over the Western Pacific and East Indian Ocean, the sampling bias is not reduced by using monthly mean reanalysis fields (sampled at the ozonesonde locations) in DJF and JJA. This suggests that ozone varies with time and space in a complex manner, and a dense (in both space and time) network would be required to capture the regional and seasonally representative model biases in this region. The probability distribution function (PDF) estimated using monthly mean reanalysis and model fields also differs largely between the two samplings (Fig. 10). Over the Western Pacific and East Indian Ocean in SON at 500 hPa, the multi-model mean shows a sharp peak around 54–58 ppb, in contrast to the broad distribution seen in the reanalysis with two peaks around 65 ppb and 35–45 ppb for the complete sampling (left bottom panel in Fig. 10). This information is useful to characterize model errors and for process-oriented model validation. On the other hand, the validation based on the ozonesonde sampling (left top panel) does not show any clear pattern and does not support model evaluation. Note that the influence of inter-annual variability was not considered in the analysis because the monthly climatological data were used by averaging over ten years for the models and five years for the reanalysis.

Although the variability of ozone is generally smaller in the SH than in the NH because of smaller local precursor emissions, large sampling biases exist even at SH mid- and high-latitudes due to the sparse ozonesonde network. In the SH mid latitudes, for example, the sign of the evaluated bias is opposite between the two cases at 200 hPa in DJF (-2.8 ppb in the complete sampling and +25.1 ppb in the ozonesonde sampling). In the SH high latitudes, evaluation results differ largely throughout the year in the middle troposphere. The temporal sampling bias mostly dominates the difference in the SH high latitudes in MAM and JJA, whereas the spatial sampling bias is also important in the SH mid latitudes in DJF and MAM. Based on the complete sampling, the ozone PDF is broadly distributed with a peak around 38 ppb at 500 hPa in SON at the SH high latitudes (right bottom panel in Fig. 10), while the multi-model mean underestimates high concentrations (>47 ppb) and shows a sharp peak of about 35 ppb. The PDF generated by the ozonesonde sampling does not provide a strong information on the distribution of the ozone (right top panel). These results highlight the advantage of using the reanalysis data for evaluating regional and seasonally representative model performance, and for characterizing these distributions.

Table 8 also shows the model evaluation results for four latitudinal bands at 500 hPa. The observations used are shown in bold in Table 2. The differences between the two evaluations are small in the NH extratropics (30–90° N) in all seasons, because of the relatively large number of observations. There are large differences in the tropics of both hemispheres: the ozonesonde network reveals a large negative sampling bias in the model evaluation in the NH tropics (Eq–30° N) in SON (-9 % in the complete sampling and -16 % in the ozonesonde sampling) and in the SH tropics (30° S–Eq) in MAM (-14 % and -21

%) and a large positive sampling bias in the NH tropics in JJA (-7 % and -3 %). Large sampling biases (> 60 %) also exist in the SH extratropics (90–30° S) in DJF and MAM due to the sparse ozonesonde network.

Further, ozonesonde sampling bias is evaluated for the control run and reanalysis comparisons. As summarized in Table 9, at 500 hPa, there are large differences (> 30 %) between the two evaluations in many regions, especially in the NH mid latitude regions in winter and in the tropics throughout the year, as also found in the ACCMIP models and reanalysis comparisons (Table 8). The analysis increments introduced by data assimilation vary with space and time, reflecting the changes in coverage and uncertainty of assimilated measurements as well as in model errors. Nevertheless, observational information was propagated globally and integrated with time through forecast steps during the data assimilation cycles. This is true for ozone because of its relatively long lifetime in the free troposphere. Therefore, the spatial distribution is well constrained by data assimilation, and we do not expect large variations in the reanalysis quality within each analysis region.

## 5.2 Seasonal variation

The seasonal cycle of tropospheric ozone is determined by various factors such as local photochemical production and atmospheric transport (e.g., Monks, 2000). Carslaw (2005), Bloomer et al. (2010), and Parrish et al. (2013) found multi-decadal changes in the amplitude and phase of the seasonal cycle at NH mid-latitudes. It was suggested that these changes can be attributed to changes in atmospheric transport patterns combined with spatial and temporal changes in emissions. CTMs have been used to explore the causal mechanisms; however, they failed to simulate several important features of the observed seasonal cycles (e.g., Ziemke et al., 2006; Stevenson et al., 2006; Parrish et al. 2014; Young et al., 2013). Accurate validation of the seasonal cycle is thus important for evaluating general model performance.

Table 10 compares the relative error in the seasonal amplitude obtained from the multi-mean model with that of the reanalysis for the complete and ozonesonde samplings. The evaluation based on the ozonesonde sampling results in a larger overestimation of the seasonal amplitude in the NH lower troposphere for most regions (+13.4–+63.4 % in the sonde sampling and -19.0–+40.2 % in the complete sampling). The large discrepancies can be attributed to large spatial variability in the seasonal variations of ozone and its model errors within each defined region and also the existence of short-term variability that is not completely captured by the ozonesonde sampling. For the Eastern US and Western Europe at 800 hPa, the sign of the bias is opposite between the two estimates. In contrast, at 200 hPa in the NH, results between the two evaluations are similar, suggesting spatial homogeneity in the seasonal cycle and its model errors within each region in the NH. Because the seasonal variations differ among different regions, the seasonal amplitude estimated for the entire NH extratropics (30–90° N) is largely different between the two estimates throughout the troposphere.

In the tropics, the estimated errors of the seasonal amplitude largely differ between the two samplings throughout the troposphere, suggesting that information obtained from the sparse ozonesonde network cannot be applied to characterize regional model errors in the seasonal cycle, even within the small defined area. The sampling bias in the seasonal amplitude estimated for the entire tropics is larger than 60 % throughout the troposphere both in the NH (Eq–30° N) and SH (30° S–Eq). Because of the large spatial variability, detailed validations using the chemical reanalysis (e.g., for each grid point) would be helpful. Also, in the SH high latitudes, large disagreements in the seasonal amplitude exist at 800 and 200 hPa.

## 6 Discussions

### 6.1 Reanalysis uncertainty

Although the reanalysis dataset provides comprehensive information for global model evaluations, its performance still needs to be improved, especially for the lower troposphere, as also discussed by Miyazaki et al. (2015). Performance can be improved by ingesting more datasets including meteorological sounders such as IASI (Clerbaux et al., 2009), AIRS (Chahine et al., 2006), and CrIS (Glumb et al., 2002). Application of a bias correction procedure for multiple measurements, which is common in numerical weather prediction (e.g., Dee, 2005), is needed to improve reanalysis accuracy. Recently developed retrievals with high sensitivity to the lower troposphere (e.g. Deeter et al., 2013; Fu et al., 2016) and the optimization of additional precursor emissions would be helpful to improve analysis of the lower troposphere. The relatively coarse resolution of the model could cause large differences between the simulated and observed concentrations at urban sites and may degrade the reanalysis.

The statistical information obtained from the reanalysis and the multi-model simulations can be used to suggest further developments for the models and observations. The analysis ensemble spread from EnKF can be regarded as uncertainty information about the analysis mean fields, indicating requirements for additional observational constraints. As shown in Fig. 11 (left panels), the relative reanalysis uncertainty is large over the tropical areas of the oceans at 800 hPa (>20 %), over the Southern Ocean at 500 hPa (10–20 %), and over the tropics of the Pacific Ocean and the Antarctic at 200 hPa (>16 %). Conversely, the reanalysis uncertainty is small from the tropics to mid-latitudes in both hemispheres at 500 hPa (<11 %). Miyazaki et al (2015) investigated that the analysis spread is caused by errors in the model input data, model processes, and assimilated measurements, and it is reduced if the analysis converges to a true state. The analysis spread is smaller in the extratropical lower stratosphere than in the tropical upper troposphere at 200 hPa, because of the high accuracy of the MLS measurements. In contrast, in the middle troposphere, the analysis spread is generally smaller in the tropics than the extratropics because of the higher sensitivities in the TES retrievals. Note that the data assimilation setting influences the analysis uncertainty estimation in the reanalysis. In particular, the analysis spread was found to be sensitive to the choice of ensemble size (Miyazaki et al., 2012b). A large ensemble size is essential to capture the proper background error covariance structure (i.e., analysis uncertainty).

The five-year reanalysis (2005–2009) may cause biases in the estimated model errors in the evaluation of the 2000 decade ACCMIP simulations that used decadal-averaged SST boundary conditions and biomass burning emissions averaged over 1997–2006 (Lamarque et al., 2010). It may neglect the influences of interannual and decadal changes in both anthropogenic and biomass emissions and meteorology. Longer-term reanalysis and time-consistent validation are required to obtain more robust error estimations.

### 6.2 Model uncertainty

The variability across the ensemble models (i.e., ensemble spread) identifies where the models are most consistent or uncertain (center panels in Fig. 11). As discussed by Young et al. (2013), the relative spread among the ACCMIP models is large over the tropical areas of the oceans in the lower and middle troposphere, a reflection of the important differences among the models in

various processes such as convective processes, lightning sources, biogenic emission sources with related chemistry. The large relative spread (>20%) at the NH mid-latitudes and in the SH at 200 hPa may be associated with the different representations of the tropopause and STE among models. In contrast, the relative spread is small around 20–40°N at 500 hPa (< 10 %).

The simultaneous enhancement of the analysis uncertainty (c.f., Section 6.1), together with the model spread, indicates low robustness of the validation results for some tropical regions over the oceans in the lower troposphere, and over the tropics in the Pacific Ocean as well as the Antarctic at 200 hPa. Meanwhile, the magnitudes of the model spread and analysis uncertainty differ considerably for some regions. At 200 hPa and higher in the extratropics, the analysis uncertainty is smaller than the model spread, where the reanalysis fields are strongly constrained by MLS measurements. These results suggest that further improvements, for instance, on the representation of the tropopause and STE are required for some of the models, in order for the reanalysis and ensemble models to have similar levels of uncertainty. In the lower troposphere, in contrast, the larger analysis uncertainty than the ensemble spread suggests that further observational constraints are required for the reanalysis for a fair comparison.

The ACCMIP model standard deviation with respect to the reanalysis could be used to identify the averaged uncertainty of ACCMIP models (right panels in Fig. 11). The standard deviation is large at NH high latitudes and over the tropical ocean areas at 800 hPa, over the SH tropics at 500 hPa, and in the SH extratropics at 200 hPa (> 25 %).

## 6.3  Implications into model improvements and climate studies

Numerous studies have identified decadal-scale changes in global tropospheric ozone using observations, such as the shift in the seasonal cycle at NH mid-latitudes and trends observed over many regions (e.g., Parrish et al., 2014; Cooper et al., 2014). A long-record of the reanalysis will allow detailed structures in simulated inter-annual and long-term variations to be evaluated in association with changes in human activities and natural processes. However, any discontinuities in the availability and coverage of the assimilated measurement will affect the quality of the reanalysis and estimated interannual variability, which limit the usability of a long term reanalysis for model evaluation, as discussed in Miyazaki et al (2015) for chemical reanalyses and in Thorne and Vose (2010) for climate reanalyses. This also requires a bias-correction procedure for each assimilated measurements, in order to improve the reanalysis quality (Inness et al, 2013). It is noted that the influence of ENSO was not considered in ACCMIP due to a decadal-averaged SST boundary condition, which limits the evaluation of inter-annual variations and could lead to bias in the ACCMIP models and reanalysis comparisons.

Process-oriented validations using the reanalysis would be useful for understanding the uncertainty in simulated ozone fields and associated mechanisms. The ACCMIP models reveal large variations in short-lived species such as OH and ozone precursors (Naik et al., 2013; Voulgarakis et al., 2013), whereas information obtained from direct in-situ measurements cannot be applied for investigating global distributions because of the limited coverage of the measurements and the large spatial variability of concentrations. Miyazaki et al. (2012b, 2015) demonstrated that the multiple-species assimilation results in a strong influence on both assimilated and non-assimilated species. Validation of various species using the chemical reanalysis product can be used to identify potential sources of error in the simulated ozone fields. Meanwhile, the global monthly products of precursor emissions from the chemical reanalysis calculations (Miyazaki et al., 2012a, 2014, 2016) can be used to validate

emission inventories and $LNO_x$ source parameterizations used in model simulations. As changes in tropospheric ozone burden associated with different future scenarios show a broadly linear relation to changes in $NO_x$ emissions (Stevenson et al., 2006), evaluations using up-to-date estimated emissions (Miyazaki et al., 2016) may prove useful to partly validate emissions for each scenario.

The performance of the simulated radiative forcing is largely influenced by representation of ozone in model simulations (Bowman et al., 2013; Shindell et al., 2013; Stevenson et al., 2013). Bowman et al (2013) suggested that overestimation of the OLR in the tropical seas of the east Atlantic Ocean and over Southern Africa is associated with model ozone errors, a persistent feature in all ACCMIP models, which was also found in this study using the reanalysis. Validation of short-lived species is also important for evaluating the radiative forcing because simulated OH fields influence simulated climates through

for instance their influences on methane (Voulgarakis et al., 2013). Thus, detailed information on model errors in ozone and other short-lived species could be used to improve estimates of radiative forcing in climate studies. Meanwhile, model biases for present-day ozone may be correlated with biases in other time periods. Young et al. (2013) showed that ACCMIP models with high, present day ozone burdens also had high burdens for the other periods of time, including the preindustrial period. Thus, the validation of present-day ozone fields using the reanalysis has the potential to evaluate preindustrial to present day

ozone radiative forcing.

## 7    Conclusions

We conducted a eight-year tropospheric chemistry reanalysis by assimilating multiple chemical species from the OMI, MLS, TES, MOPITT, SCIAMACHY, and GOME-2 to provide a gridded, chemically consistent estimate of concentrations and precursor emissions. This study explores the potential of atmospheric chemical reanalysis to evaluate global tropospheric ozone

of multi-model chemistry-climate model simulations. The evaluation results are also used to quantify the ozonesonde network sampling bias. Validation of the chemical reanalysis using global ozonesondes shows good agreement throughout the free troposphere and lower stratosphere for both seasonal and year-to-year variations.

The reanalysis product provides comprehensive and unique information on global ozone distributions for the entire troposphere and on the weakness of the individual models and multi-model mean. We found that the ACCMIP multi-model mean

overestimates ozone concentration in the NH extratropics throughout the troposphere (by 6–11 ppb and 800 hPa and by 2–9 ppb at 500 hPa for the zonal and annual mean concentration), and underestimates it in the SH tropics in the lower and middle troposphere by about 9 ppb over the eastern Pacific, by up to 18 ppb over the Atlantic, and by up to 8 ppb over the Indian Ocean. Most models underestimate the spatial variability of the annual mean concentration in the NH extratropics at 800 hPa (by up to 50 %) and in the SH extratropics at 800 and 500 hPa (by up to 70 %). The multi-model mean overestimates the

seasonal amplitude in the NH by 50–70 % in the lower troposphere and by 25–40 % in the middle troposphere, whereas the seasonal amplitude is underestimated by 15–25 % at 200 hPa in the NH extratropics. The seasonal amplitude in the NH extratropics shows great diversity among models. The NH/SH ratio is overestimated by 22–30 % in the free troposphere in the multi-model mean; this can be attributed to both a concentration high bias in the NH and a concentration low-bias in the SH

in most models. The performance of the ACCMIP model when compared with the reanalysis is qualitatively similar for most cases from that shown by Young et al. (2013) using the ozonesonde measurements but quantitatively different because of the ozonesonde network sampling bias.

We quantified the ozonesonde network sampling bias and how reanalysis can help extend the range of that network as a kind of "transfer standard". To estimate the sampling biases in the ACCMIP model evaluation, we compared two evaluation results of the mean model bias, using the chemical reanalysis based on the complete and ozonesonde samplings. For instance, the ozonesonde sampling bias in the evaluated model bias (relative to the model bias for the complete sampling) is largely negative (positive) in MAM (in DJF) by 40–50 % over the Western Pacific and East Indian Ocean and largely negative by 110 % in MAM over the equatorial Americas at 500 hPa. For the global tropics, the ozonesonde sampling bias is largely negative by 80 % in the NH (Eq–30° N) in SON and by 50 % in the SH (30° S–Eq) in MAM. The ozonesonde sampling bias is typically smaller than 30 % for the NH polar regions except in boreal winter and over the equatorial Americas, the Atlantic Ocean and Africa, and at the SH mid-latitudes in austral winter and spring from the lower to middle troposphere. Although the spatial and temporal variability is generally smaller in the SH than in the NH, the ozonesonde sampling bias cannot be negligible for capturing the regionally and monthly representative model errors even in the SH. Large sampling biases (> 60 %) exist in the SH extratropics (90–30° S) in DJF and MAM. The evaluation of the seasonal cycle of tropospheric ozone is also largely limited by the ozonesonde sampling bias. The evaluation based on the ozonesonde sampling introduces a larger overestimation of the seasonal amplitude than that based on the complete sampling for most of the surrounding areas in the NH lower troposphere, whereas the two estimates are largely different for the entire tropical regions. Therefore, there is an advantage of the reanalysis data for evaluating actual regionally and seasonally representative model performance required for model improvements. However, the network provides critical independent validation of the reanalysis, which can provide a much broader spatial constraint on chemistry-climate model performance.

The proposed model validation approach provides regionally and temporally representative model performance; this could ensure more accurate predictions for the chemistry–climate system. In future studies, validation of multiple species concentrations and precursor emissions from reanalysis would be useful in identifying error sources in model simulations. In particular, the response of tropospheric composition to changing emissions over decadal time scales is still not captured in CCMs relative to a few remote sites (Parrish et al, 2014). Recent increases in emissions from China have been linked to changes in tropospheric ozone concentrations (Verstraeten et al, 2015). Over the next decade, a new constellation of of low Earth Orbiting sounders, e.g., IASI, AIRS, CrIS, Sentinel-5p (TROPOMI), Sentinel-5 and geostationary satellites (Sentinel-4, GEMS, and TEMPO) will provide even more detailed knowledge of ozone and its precursors (Bowman, 2013). Assimilating these datasets into a decadal chemical reanalysis will be a more direct means of quantifying the response of atmospheric composition to emissions at climate relevant time scales, which should be a more direct test on chemistry-climate change scenarios. Combining many observations requires a bias correction procedure for each assimilated measurement to improve the reanalysis quality but needs to be carefully checked. In order for reanalysis to be more effective in evaluating performance, chemistry-climate model simulations that represent year-specific ozone distributions over the contemporary period are urgently needed. We also

plan to apply the proposed evaluation approach to a more recent model inter-comparison project, the Chemistry-Climate Model Initiative (CCMI).

*Acknowledgements.* We acknowledge the use of data products from the NASA AURA and EOS Terra satellite missions. We also acknowledge the free use of tropospheric $NO_2$ column data from the SCIAMACHY, GOME-2, and OMI sensors from www.temis.nl. We would also like to thank the editor and two anonymous reviewers for their valuable comments. This work was supported through JSPS KAKENHI grant numbers 15K05296, 26220101, and 26287117, and Coordination Funds for Promoting AeroSpace Utilization by MEXT, JAPAN.

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

**Table 1.** Measurements used for data assimilation in the chemical reanalysis.

| Sensor | Satellite | Version | Period | Species | Type | Reference |
|--------|-----------|---------|--------|---------|------|-----------|
| OMI | AURA | DOMINO v2 | 2005–2009 | $NO_2$ | Tropospheric column | Boersma et al (2011) |
| SCIAMACHY | ENVISAT | TM4NO2A v2.3 | 2005–2009 | $NO_2$ | Tropospheric column | Boersma et al (2004) |
| GOME-2 | MetOp-A | TM4NO2A v2.3 | 2007–2009 | $NO_2$ | Tropospheric column | Boersma et al (2004) |
| TES | AURA | v5 | 2005–2009 | Ozone | Profile | Herman and Kulawik (2013) |
| MLS | AURA | v4.2 | 2005–2009 | Ozone/$HNO_3$ | Profile above 215/150 hPa | Livesey et al. (2011) |
| MOPITT | TERRA | v6 TIR | 2005–2009 | CO | Profile | Deeter et al. (2013) |

Table 2: Ozonesonde observation sites used in this study. All the data are used for the evaluation of reanalysis data (Section 3), whereas selected observations (shown in bold) based on the compilation by Tilmes et al. (2012) are used for the evaluation of ACCMIP models and to investigate ozonesonde sampling biases (Section 4 and 5).

| Station | Latitude | Longitude | Period | Profiles |
|---------|----------|-----------|--------|----------|
| **Alert** | 82.5 | -62.3 | 2005-2009 | 259 |
| **Eureka** | 80.0 | -85.9 | 2005-2009 | 423 |
| **Ny Alesund** | 78.9 | 11.9 | 2005-2009 | 382 |
| **Resolute** | 74.7 | -95 | 2005-2009 | 205 |
| Summit | 72.6 | -38.5 | 2008 | 36 |
| Barrow | 71.3 | -156.6 | 2008 | 27 |
| Sodankyla | 67.4 | 65.2 | 2005-2006 | 161 |
| **Lerwich** | 60.1 | -1.2 | 2005-2009 | 253 |
| **Churchill** | 58.7 | 94.1 | 2005-2009 | 214 |
| **Edmonton** | 53.6 | -114.1 | 2005-2009 | 265 |
| **Goose Bay** | 53.3 | -60.4 | 2005-2009 | 246 |
| **Legionowo** | 52.4 | 21.0 | 2005-2009 | 284 |
| **Lindenberg** | 52.2 | 14.1 | 2005-2009 | 276 |
| **De Bilt** | 52.1 | 5.2 | 2005-2009 | 314 |
| Valentia Observatory | 51.9 | -10.3 | 2005-2009 | 202 |
| **Uccle** | 50.8 | 4.4 | 2005-2009 | 724 |
| Bratt's Lake | 50.2 | -104.7 | 2005-2009 | 263 |
| **Praha** | 50.0 | 14.4 | 2005-2009 | 289 |
| Kelowna | 49.9 | -119.4 | 2005-2009 | 285 |
| **Hohenpeissenberg** | 47.8 | 11.0 | 2005-2009 | 635 |

Table 2: Ozonesonde observation sites used in this study. All the data are used for the evaluation of reanalysis data (Section 3), whereas selected observations (shown in bold) based on the compilation by Tilmes et al. (2012) are used for the evaluation of ACCMIP models and to investigate ozonesonde sampling biases (Section 4 and 5).

| Station | Latitude | Longitude | Period | Profiles |
|---|---|---|---|---|
| **Payerne** | 46.5 | 6.6 | 2005 | 774 |
| Richland | 46.2 | -119.2 | 2006 | 24 |
| Egbert | 44.2 | -79.8 | 2005-2009 | 231 |
| Sable Island | 44.0 | -59.9 | 2006 | 28 |
| Yarmouth | 43.9 | -66.1 | 2005-2009 | 213 |
| Paradox | 43.9 | -73.6 | 2006 | 8 |
| Sapporo | 43.1 | 141.3 | 2005-2009 | 206 |
| Walsingham | 42.6 | -80.6 | 2006 | 43 |
| Narragansett | 41.5 | -71.3 | 2006, 2008 | 51 |
| Valparaiso | 41.5 | -87.0 | 2006 | 18 |
| Trinidad Head | 40.8 | -124.2 | 2006, 2008 | 83 |
| Barajas | 40.5 | -3.6 | 2005-2009 | 268 |
| Ankara | 40.0 | 32.9 | 2005-2009 | 101 |
| Beltsville | 39.0 | -76.5 | 2006 | 12 |
| **Wallops Island** | 37.9 | -75.5 | 2005-2009 | 283 |
| Tateno | 36.1 | 140.1 | 2005-2009 | 232 |
| **Huntsville** | 35.3 | -86.6 | 2005-2007 | 162 |
| Table Mountain | 34.4 | -117.7 | 2006 | 44 |
| Holtville | 32.8 | -115.4 | 2006 | 13 |
| Isfahan | 32.5 | 51.7 | 2005-2009 | 57 |
| Houston | 29.7 | -95.3 | 2006 | 36 |
| Dehli | 28.3 | 1.3 | 2006, 2007, 2009 | 54 |
| **Naha** | 26.2 | 127.7 | 2005 | 198 |
| **Hong Kong** | 22.3 | 114.2 | 2005-2009 | 237 |
| Ha Noi | 21.0 | 105.8 | 2005-2009 | 174 |
| **Hilo** | 19.7 | -155.1 | 2005-2009 | 240 |
| Tecamec | 19.3 | -99.2 | 2006 | 35 |
| Barbados | 13.2 | -59.5 | 2006 | 27 |
| Poona | 18.6 | 73.9 | 2007-2009 | 28 |
| Heredia | 10.0 | -84.1 | 2005-2007 | 82 |

Table 2: Ozonesonde observation sites used in this study. All the data are used for the evaluation of reanalysis data (Section 3), whereas selected observations (shown in bold) based on the compilation by Tilmes et al. (2012) are used for the evaluation of ACCMIP models and to investigate ozonesonde sampling biases (Section 4 and 5).

| Station | Latitude | Longitude | Period | Profiles |
|---|---|---|---|---|
| Thiruvananthapuram | 8.5 | 77.6 | 2006-2009 | 102 |
| Cotonou | 6.2 | 2.2 | 2005-2007 | 97 |
| **Paramaribo** | 5.8 | -55.2 | 2005-2009 | 312 |
| Kuala Lampur | 2.7 | 101.7 | 2005-2009 | 146 |
| **San Cristobal** | -0.9 | -89.6 | 2005-2008 | 131 |
| **Nairobi** | -1.3 | 36.8 | 2005-2009 | 190 |
| Malindi | -3.0 | 40.2 | 2005-2006 | 19 |
| Natal | -5.8 | -35.2 | 2005-2009 | 227 |
| **Watukosek** | -7.5 | 112.6 | 2005-2009 | 98 |
| Ascension Island | -8.0 | -14.4 | 2005-2009 | 269 |
| **American Samoa** | -14.2 | -170.6 | 2005-2009 | 130 |
| **Fuji** | -18.1 | 178.4 | 2005, 2007-2009 | 57 |
| Reunion Island | -21.1 | 55.5 | 2005-2009 | 236 |
| Pretoria | -25.9 | 28.2 | 2005-2007 | 95 |
| Broadmeadows | -37.7 | 145.0 | 2005-2009 | 231 |
| **Lauder** | -45.0 | 169.6 | 2005-2009 | 282 |
| **Macquarie Island** | -54.5 | 158.9 | 2005-2009 | 214 |
| Ushuaia | -54.9 | -68.3 | 2008-2009 | 60 |
| **Marambio** | -64.2 | -56.7 | 2005-2009 | 358 |
| Davis | -68.6 | 78.0 | 2005-2009 | 120 |
| **Syowa** | -69.0 | 39.6 | 2005-2009 | 236 |
| Maitri | -70.5 | 11.4 | 2005-2008 | 47 |
| **Neumayer** | -70. 6 | -8.3 | 2005-2009 | 383 |

**Table 3.** Chemical reanalysis (or control run in brackets) minus ozonesonde comparisons of mean ozone concentrations in 2005–2009. RMSE is the root-mean-square error. Units of bias and RMSE are ppb. T-Corr is the temporal correlation.

| | 90–30° S | | | 30° S–Eq | | | Eq-30° N | | | 30–90° N | | |
|---|---|---|---|---|---|---|---|---|---|---|---|---|
| | Bias | RMSE | T-Corr | Bias | RMSE | T-Corr | Bias | RMSE | T-Corr | Bias | RMSE | T-Corr |
| 850–500 | -0.6 | 4.3 | 0.88 | 2.4 | 6.8 | 0.96 | 2.6 | 7.4 | 0.81 | -1.5 | 6.3 | 0.90 |
| hPa | (-2.3) | (4.9) | (0.93) | (1.5) | (7.3) | (0.87) | (2.6) | (7.9) | (0.69) | (-3.3) | (6.8) | (0.92) |
| 500–200 | 0.1 | 16.5 | 0.88 | 0.5 | 8.5 | 0.95 | 1.3 | 9.8 | 0.78 | -4.1 | 23.2 | 0.98 |
| hPa | (32.5) | (33.4) | (0.78) | (1.8) | (10.4) | (0.82) | (4.2) | (12.3) | (0.67) | (20.1) | (31.7) | (0.92) ) |
| 200–90 | 29.8 | 77.1 | 0.98 | 4.2 | 18.5 | 0.93 | 2.8 | 27.8 | 0.83 | 8.9 | 85.7 | 0.99 |
| hPa | (365.6) | (277.8) | (0.84) | (60.3) | (58.0) | (0.82) | (60.0) | (62.6) | (0.86) | (260.3) | (209.7) | (0.98) |

**Table 4.** ACCMIP model mean minus reanalysis comparisons of the mean ozone concentrations. Units of bias and RMSE are ppb. S-Corr is the spatial correlation coefficient.

| | 90–30° S | | | 30° S–Eq | | | Eq–30° N | | | 30–90° N | | |
|---|---|---|---|---|---|---|---|---|---|---|---|---|
| | Bias | RMSE | S-Corr | Bias | RMSE | S-Corr | Bias | RMSE | S-Corr | Bias | RMSE | S-Corr |
| 800 hPa | 0.0 | 2.0 | 0.99 | -3.2 | 4.5 | 0.94 | 2.4 | 3.5 | 0.97 | 7.6 | 7.9 | 0.97 |
| 500 hPa | -3.5 | 4.0 | 0.99 | -7.0 | 7.7 | 0.95 | -3.1 | 4.1 | 0.96 | 4.7 | 5.4 | 0.57 |
| 200 hPa | -20.7 | 23.9 | 0.99 | -2.0 | 4.2 | 0.99 | -0.5 | 2.9 | 0.99 | -15.7 | 20.1 | 1.00 |

**Table 5.** ACCMIP models minus reanalysis comparisons of the mean ozone concentrations at 500 hPa. Units of bias are ppb.

| | 90–30° S | | 30° S–Eq | | Eq–30° N | | 30–90° N | |
|---|---|---|---|---|---|---|---|---|
| | Bias | S-Corr | Bias | S-Corr | Bias | S-Corr | Bias | S-Corr |
| 1. CESM-CAM | -8.5 | 0.99 | -14.6 | 0.90 | -9.5 | 0.91 | 4.8 | 0.12 |
| 2. CICERO-OsloCTM2 | -11.4 | 0.99 | -8.1 | 0.89 | -6.7 | 0.94 | -5.9 | 0.82 |
| 3. CMAM | -0.8 | 0.99 | -7.7 | 0.85 | -6.6 | 0.91 | -0.5 | 0.73 |
| 4. EMAC | 0.5 | 0.96 | 2.0 | 0.77 | 3.5 | 0.90 | 1.6 | 0.87 |
| 5. GEOSCCM | -2.1 | 0.98 | -7.4 | 0.90 | -4.7 | 0.91 | 5.2 | 0.59 |
| 6. GFDL-AM3 | 4.4 | 0.99 | -2.4 | 0.95 | 0.9 | 0.95 | 8.0 | 0.03 |
| 7. GISS-E2-R | -0.2 | 0.98 | -3.6 | 0.87 | 1.0 | 0.91 | 14.5 | 0.04 |
| 8. GISS-E2-TOMAS | 5.9 | 0.96 | 1.3 | 0.83 | 4.6 | 0.90 | 17.2 | -0.07 |
| 9. HadGEM2 | -6.6 | 0.98 | -12.8 | 0.91 | -7.8 | 0.90 | -0.8 | 0.86 |
| 10. LMDzORINCA | -5.0 | 0.98 | -6.5 | 0.94 | -4.6 | 0.95 | 2.8 | 0.60 |
| 11. MIROC-CHEM | -4.2 | 0.98 | -0.5 | 0.92 | 1.3 | 0.93 | -2.0 | 0.87 |
| 12. MOCAGE | -9.6 | 0.93 | -12.2 | 0.47 | -4.0 | 0.82 | 11.8 | -0.11 |
| 13. NCAR-CAM3.5 | -4.0 | 0.99 | -10.4 | 0.93 | -4.3 | 0.93 | 4.3 | 0.45 |
| 14. STOC-HadAM3 | -8.7 | 0.96 | -8.9 | 0.86 | -4.0 | 0.94 | 2.4 | 0.38 |
| 15. UM-CAM | -2.8 | 0.96 | -13.4 | 0.85 | -5.8 | 0.85 | 7.5 | 0.79 |

**Table 6.** Regions and observation sites used in model evaluation in Section 5. The 11 regions are defined following Tilmes et al. (2012). See also Fig. 8.

| Region | Station (Lat/Lon) |
|---|---|
| NH polar West (60° N–90° N, 120° W–40° W) | Alert (83/-62), Eureka (80/-86), Resolute (74/-95) |
| NH polar East (60° N–90° N, 40° W–30° E) | NyAlesund (79/12), Lerwick (60/-1) |
| Canada (53° N–60° N, 120° W–50° W) | Churchill (59/-94), Edmonton (53/-114), Goosebay (53/-60) |
| Western Europe (45° N–55° N, 0° E–24° E) | Legionowo (52/21), Lindenberg (52/14), Debilt (52/5), Uccle (51/4), Praha (50/15), Hohenpbg (48/11), Payerne (47/7) |
| Eastern US (32° N–38° N, 129° E–142° W) | Wallops Island (38/-76), Huntsville (35/-87) |
| NH subtropics (15° N–29° N, 110° E–150° W) | Hilo (19/-155), Hongkong (22/114), Naha (26/128) |
| W. Pacific/E. Indian (20° S–6° S, 110° E–160° W) | Fiji (-18/178), Watukosek (-8/113), Samoa (-14/-171) |
| Equatorial Americas (4° S–9° N, 100° W–45° W) | Paramaribo (6/-55), Sancristobal (-1,-90) |
| Atlantic/Africa (11° S–2° N, 40° W–40° E) | Nairobi (-1/37), Natal (-5/-35), Ascension (-8/-14) |
| SH mid latitudes (60° S–40° S, all longitudes) | Lauder (-45/170), Macquarie (-55/159) |
| SH high latitudes (60° N–80° N, all longitudes) | Marambio (-64/-57), Syowa (-69/40), Neumayer (-71/-8) |

**Table 7.** The reanalysis ozone concentration differences between the ozonesonde sampling (for both time and space using two-hourly reanalysis fields) and the complete sampling at 500 hPa (in % relative to the complete sampling). Results using monthly reanalysis fields sampled at the ozonesonde locations are also shown in brackets.

| | DJF | MAM | JJA | SON |
|---|---|---|---|---|
| NH polar West | -1.2 (-1.8) | 0.2 (-1.5) | 2.1 (-1.1) | 0.8 (-2.5) |
| NH polar East | 1.8 (0.9) | 2.8 (1.2) | 2.8 (1.6) | 2.9 (1.0) |
| Canada | -0.1 (0.0) | 2.4 (-0.5) | 1.5 (-0.3) | 0.3 (0.3) |
| Western Europe | -0.3 (-0.1) | 1.5 (-0.2) | 1.0 (-0.4) | 1.8 (-0.1) |
| Eastern US | -0.2 (-0.1) | 4.4 (0.7) | 13.8 (3.3) | 4.4 (0.6) |
| NH subtropics | 5.9 (3.8) | 6.1 (3.0) | 4.0 (4.5) | 12.3 (7.8) |
| W. Pacific/E. Indian | 4.5 (-9.8) | 5.0 (-6.0) | -3.2 (-9.4) | 2.2 (-13.9) |
| Equatorial Americas | 4.3 (2.1) | 7.8 (5.5) | 2.0 (0.0) | -0.0 (-1.8) |
| Atlantic/Africa | 7.5 (4.7) | 0.7 (0.7) | -3.8 (-2.6) | 3.7 (1.2) |
| SH mid latitudes | 0.3 (2.5) | 3.9 (2.5) | 2.0 (1.9) | 2.1 (2.0) |
| SH high latitudes | 0.8 (-2.4) | 4.2 (-0.9) | 3.4 (-0.4) | 3.9 (-0.7) |
| 30–90° N | -0.9 (-0.4) | 1.0 (-0.3) | 1.0 (-2.6) | 0.5 (-0.9) |
| Eq–30° N | 8.6 (7.2) | 12.2 (9.8) | 0.4 (0.1) | 13.3 (8.9) |
| 30° S–Eq | 8.1 (0.7) | 6.2 (1.5) | 1.5 (-0.3) | 4.8 (-2.7) |
| 90–30° S | -13.0 (-13.6) | -6.0 (-9.1) | -5.7 (-7.7) | -9.3 (-11.5) |

**Table 8.** Median of the ACCMIP models minus reanalysis at 500 hPa (in % relative to the reanalysis concentrations). Results presented include the regional averages (Regional), for the ozonesonde temporal/spatial sampling using two-hourly reanalysis fields (Sonde), and for the ozonesonde spatial sampling using monthly reanalysis fields (in brackets). Relative differences between the two estimates larger than 30 % are shown in bold.

| | DJF | | MAM | | JJA | | SON | |
|---|---|---|---|---|---|---|---|---|
| | Regional | Sonde | Regional | Sonde | Regional | Sonde | Regional | Sonde |
| NH polar West | 12.6 | 13.4 (14.1) | 13.4 | 14.3 (15.9) | 15.5 | 16.2 (19.4) | 17.5 | 15.4 (18.8) |
| NH polar East | **5.9** | **1.9 (3.0)** | 12.3 | 10.0 (11.6) | 16.5 | 13.8 (14.9) | 17.7 | 14.6 (16.6) |
| Canada | **6.7** | **5.0 (4.8)** | **7.2** | **3.9** (5.9) | **7.9** | **5.6** (7.3) | 12.9 | 12.7 (12.7) |
| Western Europe | **3.1** | **1.3 (1.1)** | **6.1** | **4.4** (6.0) | 3.0 | 2.4 (3.7) | 8.6 | 7.0 (8.9) |
| Eastern US | 3.9 | 4.6 (**2.8**) | 1.6 | **3.0 (2.5)** | 1.4 | **-0.3** (1.6) | 3.8 | **2.9** (4.1) |
| NH subtropics | -9.5 | -10.3 (-8.3) | -9.0 | -10.3 (-7.2) | **-5.5** | **-0.3 (-0.7)** | **-3.5** | **-8.6** (-4.2) |
| W. Pacific/E. Indian | **-27.3** | **-16.3 (-1.9)** | **-16.1** | **-23.4** (-12.4) | **-12.1** | -12.6 (**-6.3**) | **-16.3** | **-27.4** (-11.2) |
| Equatorial Americas | -10.9 | -9.9 (**-7.7**) | **-4.5** | **-9.6 (-7.3)** | -15.6 | -19.6 (-17.6) | -21.9 | -24.4 (-22.6) |
| Atlantic/Africa | -26.6 | -23.8 (-21.1) | -17.8 | -17.5 (-17.5) | -15.7 | -18.2 (-19.4) | -23.6 | -25.7 (-23.2) |
| SH mid latitudes | **-12.1** | **-3.6 (-6.0)** | **-6.2** | **-10.7 (-9.3)** | -12.7 | -13.3 (-13.2) | -16.8 | -17.6 (-17.6) |
| SH high latitudes | **-10.3** | **-4.3 (-1.1)** | 2.2 | **-3.4** (1.7) | **-3.6** | **-9.2 (-5.4)** | **-11.7** | **-4.7** (-11.6) |
| 30–90° N | **3.8** | 3.0 (**2.5**) | **4.7** | 5.6 (**6.9**) | 5.8 | 5.8 (6.7) | 10.0 | 10.7 (12.1) |
| Eq–30° N | -10.9 | -10.7 (-9.4) | **-10.2** | -9.8 (**-7.4**) | **-6.7** | **-2.8 (-2.6)** | **-8.7** | **-16.0** (-11.6) |
| 30° S–Eq | -16.9 | -21.8 (-14.5) | **-13.7** | **-20.8** (-15.3) | -20.1 | -18.3 (-16.5) | -20.9 | -25.2 (-17.8) |
| 90–30° S | **-10.7** | **-4.8 (-4.2)** | **-4.1** | **-6.7** (-3.6) | -11.3 | -10.3 (-8.3) | -15.3 | -14.8 (-12.6) |

**Table 9.** The control run minus reanalysis comparison of the mean ozone concentration at 500 hPa (in % relative to the reanalysis concentrations). Results are the regional averages (Regional) and at the ozonesonde temporal/spatial sampling (Sonde). Relative differences between the two estimates larger than 30 % are shown in bold.

| | DJF | | MAM | | JJA | | SON | |
|---|---|---|---|---|---|---|---|---|
| | Regional | Sonde | Regional | Sonde | Regional | Sonde | Regional | Sonde |
| NH polar West | -5.2 | -4.2 | -4.6 | -3.6 | **-3.4** | **-2.2** | -2.8 | -2.9 |
| NH polar East | -5.2 | -4.6 | -4.4 | -4.4 | -3.9 | -4.1 | **-2.1** | **-1.1** |
| Canada | **-4.1** | **-0.6** | -6.0 | -6.0 | -6.6 | -6.3 | -3.2 | -2.6 |
| Western Europe | **-2.5** | **-1.8** | -4.8 | -4.9 | -4.3 | -3.4 | 0.6 | 0.8 |
| Eastern US | **0.2** | **-3.8** | **-2.5** | **-1.9** | **1.0** | **-1.0** | **5.3** | **7.5** |
| NH subtropics | **-3.1** | **2.3** | **-0.7** | **0.8** | 6.3 | 6.6 | 7.0 | 7.3 |
| W. Pacific/E. Indian | **14.6** | **6.4** | **0.5** | **-1.2** | **3.9** | **1.4** | **-5.2** | **-6.8** |
| Equatrial Americas | **7.2** | **4.2** | **4.5** | **5.9** | **-3.4** | **-5.2** | -7.4 | -6.7 |
| Atlantic/Africa | 3.0 | 2.7 | **-5.2** | **-1.6** | **-3.7** | **-0.7** | -10.1 | -7.8 |
| SH mid latitudes | **-5.6** | **-3.7** | 0.5 | 3.4 | 4.1 | 4.5 | **-2.5** | **-0.4** |
| SH high latitudes | **-9.4** | **-7.6** | -9.6 | -7.7 | **-1.2** | **0.2** | -6.0 | -5.3 |
| 30–90° N | -2.3 | -2.3 | -3.7 | -4.2 | **-2.2** | **-2.9** | 1.3 | 0.6 |
| Eq–30° N | **1.4** | **3.8** | **-0.7** | **1.0** | **1.5** | **2.9** | 3.0 | 4.0 |
| 30° S–Eq | **6.1** | **1.6** | **2.9** | **0.3** | **6.0** | **-0.1** | -3.5 | -7.3 |
| 90–30° S | **-2.9** | **-6.0** | 1.5 | -3.2 | 5.2 | 1.9 | -1.8 | -3.3 |

**Table 10.** ACCMIP multi-model mean minus reanalysis comparisons of the seasonal amplitude of regional mean ozone concentration (in %) for the regional average (Regional) and at the ozonesonde sampling (Sonde). The seasonal amplitude is estimated as a difference between maximum and minimum monthly mean concentrations.

| | 800 hPa | | 500 hPa | | 200 hPa | |
|---|---|---|---|---|---|---|
| | Regional | Sonde | Regional | Sonde | Regional | Sonde |
| NH polar West | 40.2 | 52.0 | -7.0 | 4.5 | -24.2 | -20.9 |
| NH polar East | 14.2 | 13.4 | 10.5 | 9.4 | -16.3 | -24.2 |
| Canada | 1.0 | 27.1 | -11.2 | -22.9 | -22.1 | -14.6 |
| Western Europe | -12.2 | 38.0 | 1.0 | 7.1 | -18.5 | -18.6 |
| Eastern US | -19.0 | 71.3 | -8.6 | -13.3 | -11.1 | -15.7 |
| NH subtropics | 10.5 | 63.4 | -27.0 | 16.7 | -48.2 | -46.0 |
| W. Pacific/E. Indian | 10.5 | -42.9 | -14.3 | -24.2 | -16.3 | -33.6 |
| Equatorial Americas | -27.3 | -5.4 | -64.1 | -23.9 | -37.2 | -76.5 |
| Atlantic/Africa | -14.7 | -1.3 | -13.1 | 3.3 | -32.6 | -15.2 |
| SH mid latitudes | 7.8 | -1.5 | -45.3 | -47.5 | -40.2 | -32.6 |
| SH high latitudes | 40.0 | 4.6 | -31.1 | -36.3 | 83.6 | -20.8 |
| 30–90° N | 2.8 | 16.3 | -39.6 | 0.0 | -42.2 | -17.1 |
| Eq–30° N | 37.0 | 106.1 | -20.1 | 23.9 | -13.6 | -47.9 |
| 30° S–Eq | -16.4 | -28.5 | -9.5 | -22.9 | -36.8 | -21.8 |
| 90–30° S | 5.5 | 12.5 | -5.5 | 33.9 | -18.8 | -13.7 |

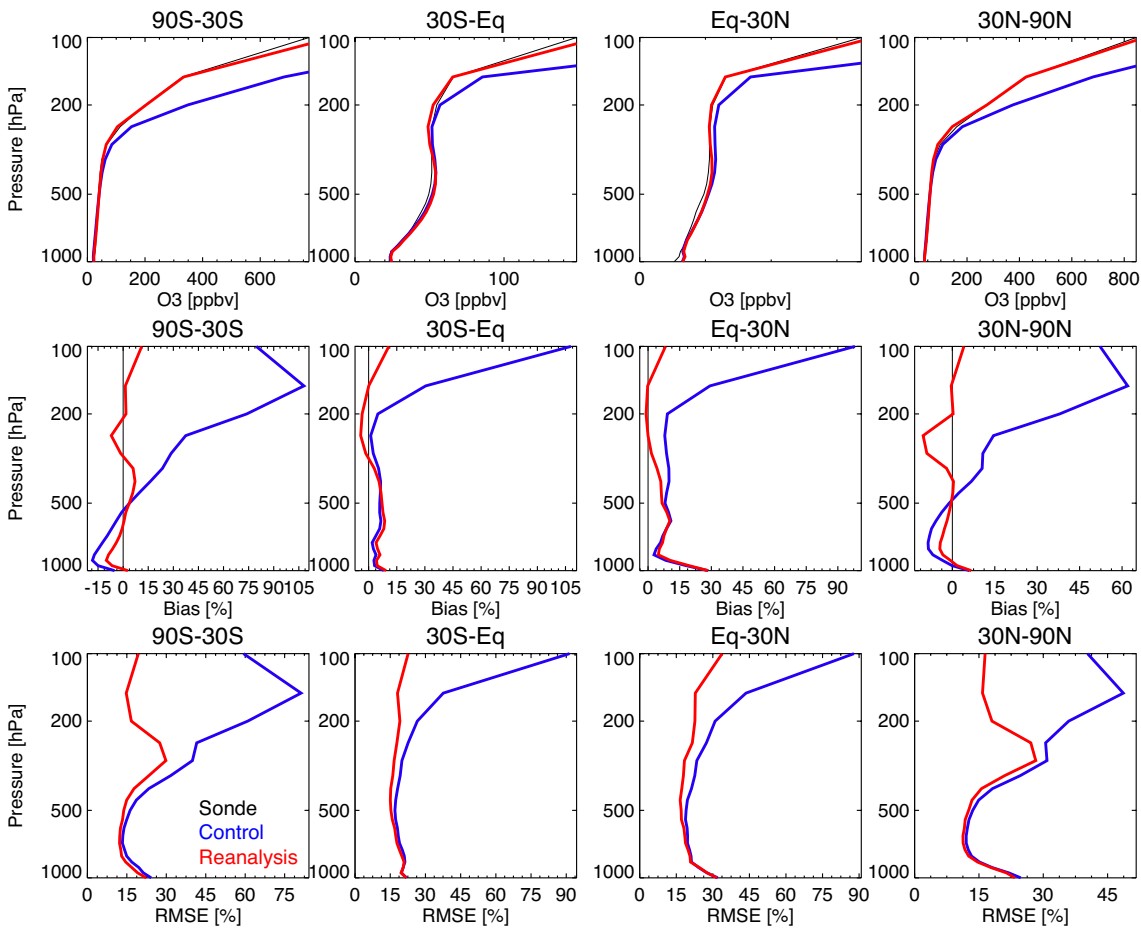

**Figure 1.** Comparison of vertical ozone profiles from ozonesondes (black), control run (blue), and reanalysis (red) averaged for the period 2005–2009. Top row shows mean profile; middle and bottom rows show mean difference and RMSE between control run and observations (blue) and between the reanalysis and the observations (red) relative to the mean ozonesonde concentrations (in %). From left to right, results are shown for SH extratropics (30–90°S), SH tropics (30°S–Eq), NH tropics (Eq–30°N), and NH extratropics (30–90°N). All ozonesonde observations taken from the WOUDC database were used in the comparison.

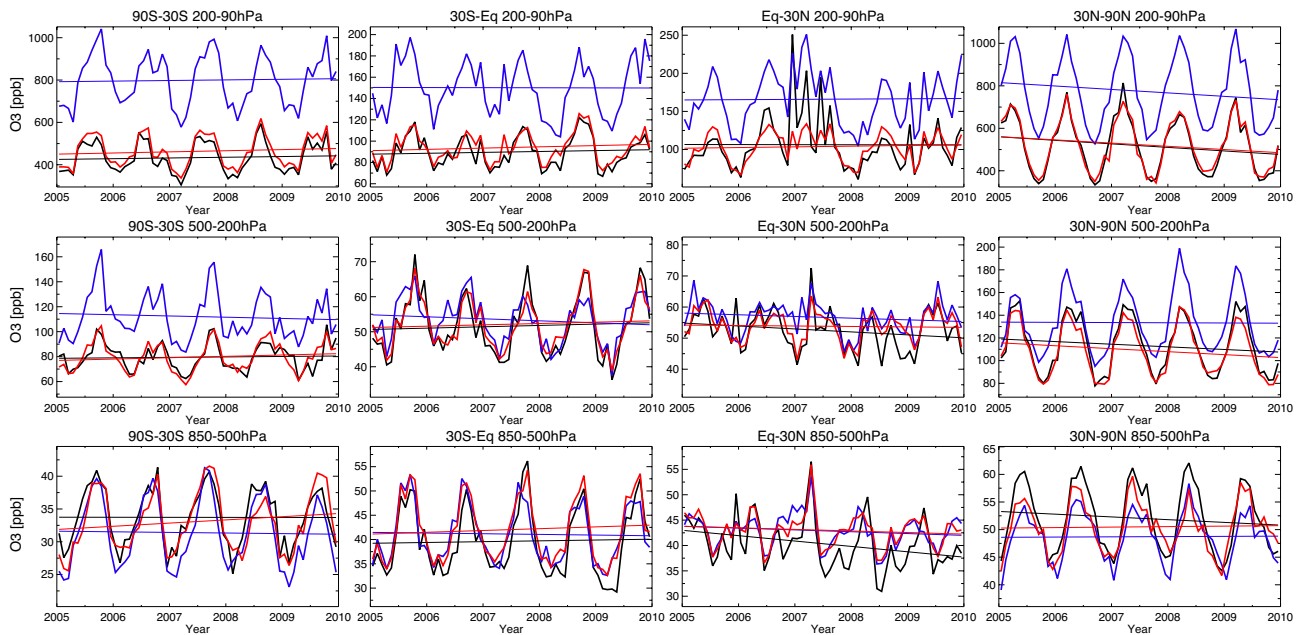

**Figure 2.** Time series of monthly mean ozone concentrations obtained from ozonesondes (black), control run (blue), and reanalysis (red) averaged between 850 and 500 hPa (top), 500 and 200 hPa (middle), and 200 and 90 hPa (bottom) for 2005–2009. From left to right the results are shown for SH extratropics (30–90°S), SH tropics (30°S–Eq), NH tropics (Eq–30°N), and NH extratropics (30–90°N).

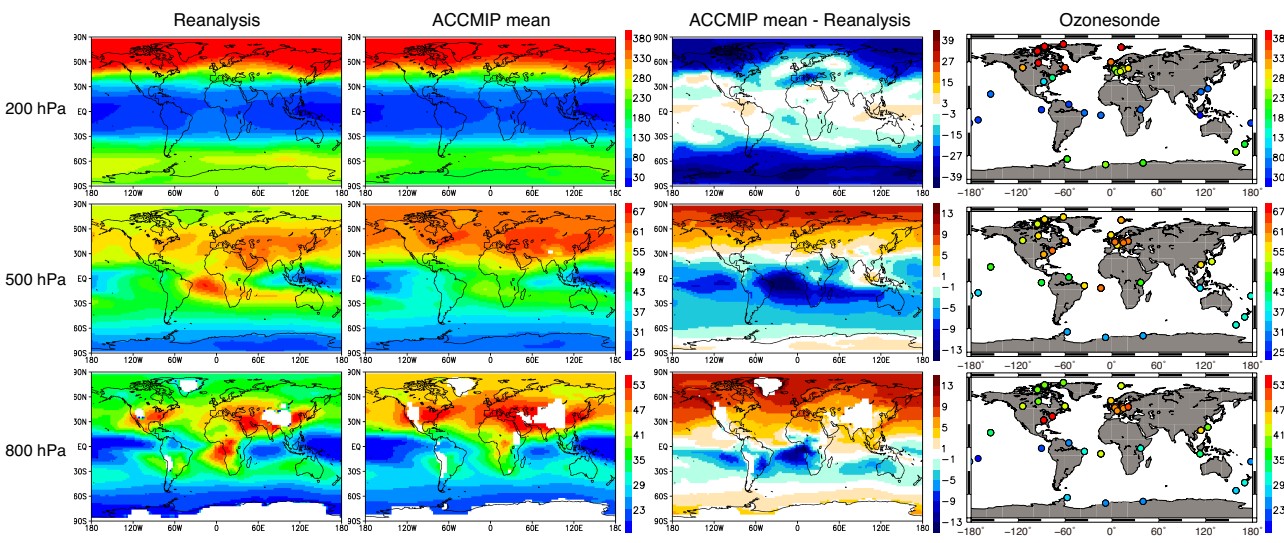

**Figure 3.** Global distributions of annual mean ozone concentrations obtained from reanalysis (left), ACCMIP model mean (2nd left), difference between ACCMIP model mean and reanalysis (3rd left), and the ozonesonde measurements used for the evaluation of ACCMIP models and ozonesonde sampling biases (right). From top to bottom, results are shown for global distributions at 200 hPa, 500 hPa, and 800 hPa. Units are ppb.

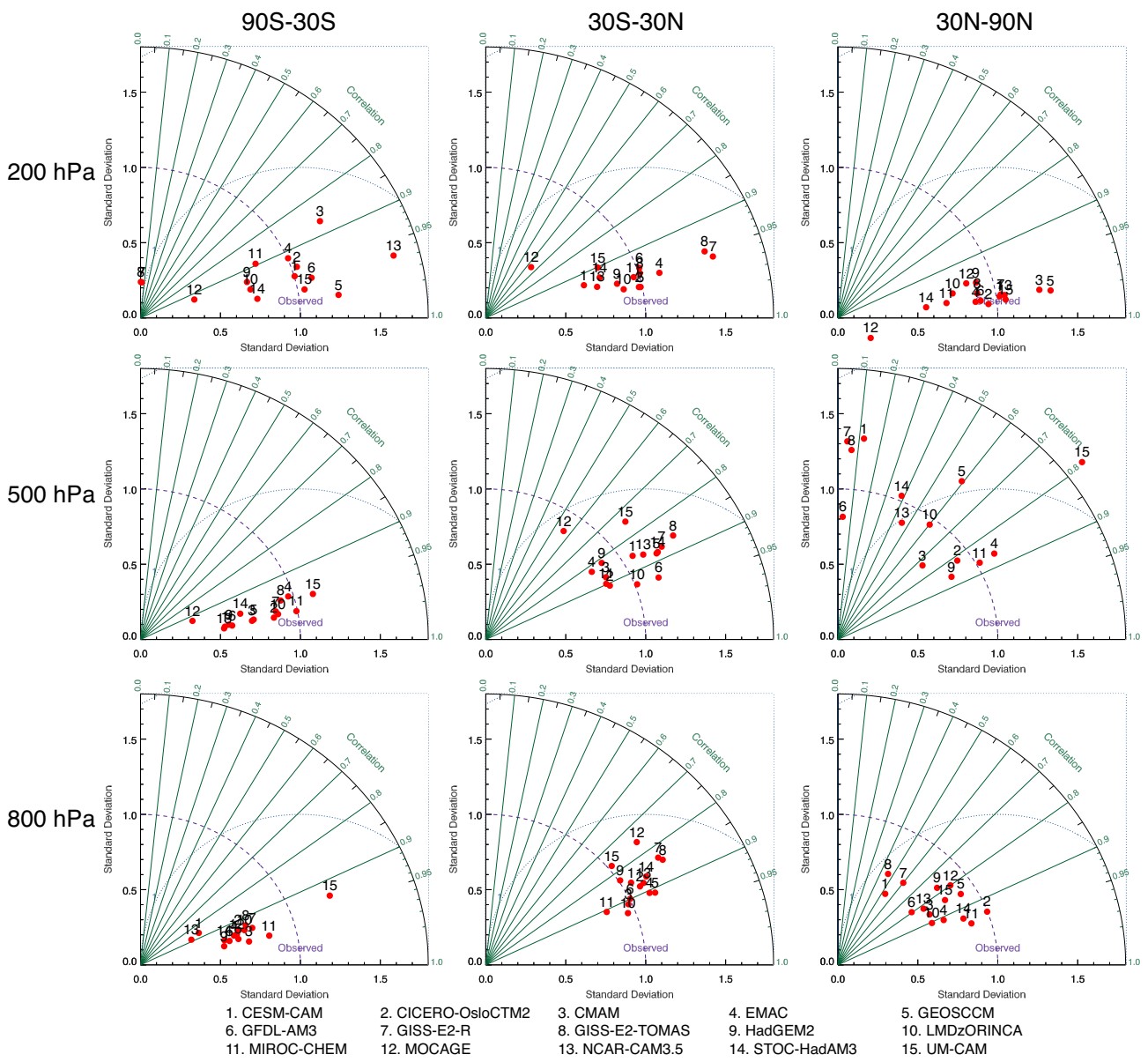

**Figure 4.** Taylor diagrams showing standard deviation normalized with respect to that of reanalysis (x-axis) and spatial correlation coefficient (y-axis) for the comparison of annual mean ozone concentrations between ACCMIP models and reanalysis for SH extratropics (90°S–30°S, left), tropics and subtropics (30°S–30°N, center), and NH extratropics (30°N–90°N, right) at 200 hPa (top), 500 hPa (middle), and 800 hPa (bottom).

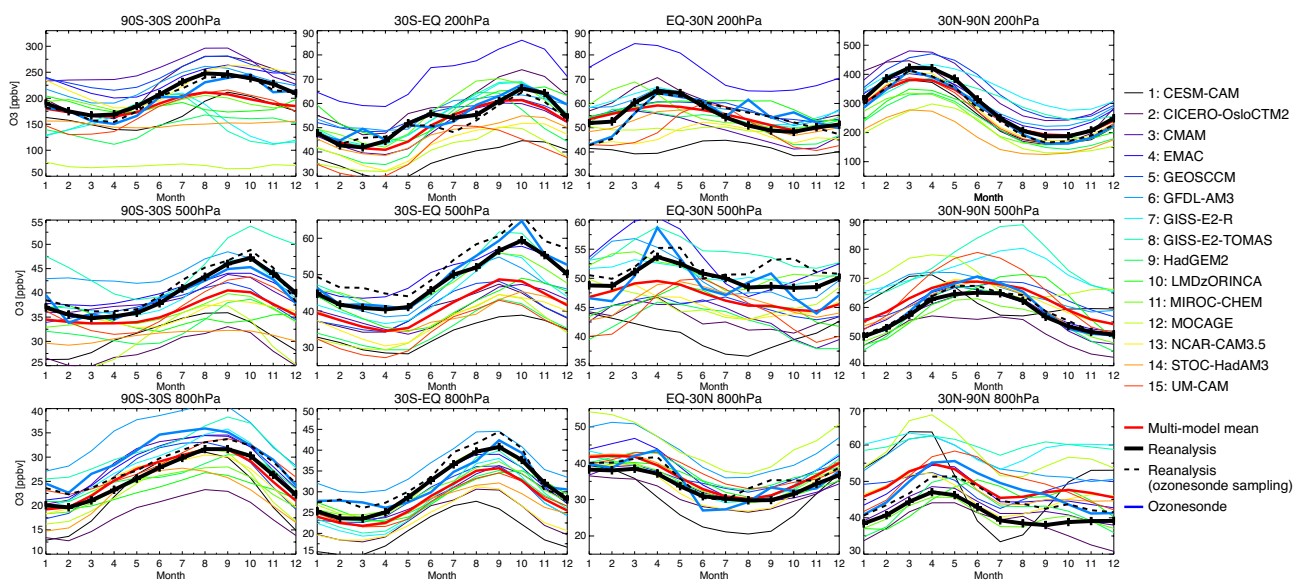

**Figure 5.** Comparison of seasonal variation of ozone concentration between the reanalysis (black lines), individual ACCMIP models (thin colored lines), ACCMIP ensemble mean (red solid line), and ozonesonde observations (blue solid line) averaged between 90°S–30°S (1st column from left), 30°S–Eq (2nd column), Eq–30°N (3rd column), and 30°N–90°N (4th column). From top to bottom, results are shown for concentrations at 200 hPa, 500 hPa, and 800 hPa. Individual model results are shown by colored thin lines. The reanalysis result is shown for the average over all model grid points (black solid line) and over the ozonesonde sampling sites/time (black dashed line). The ACCMIP model results are shown for the average over all model grid points.

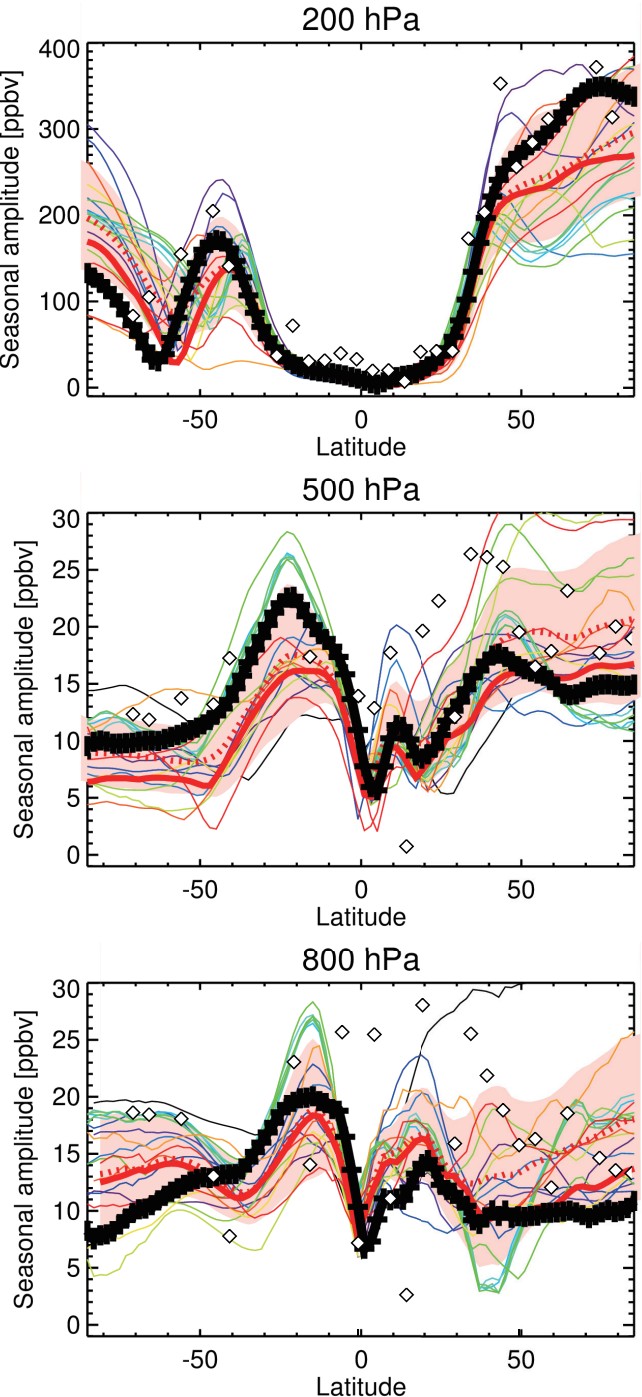

**Figure 6.** Seasonal amplitude (peak-to-peak difference based on monthly data) estimated from the reanalysis (black solid line) and ACCMIP models (thin colored lines). The $\pm 1\sigma$ deviation among ACCMIP models (i.e., model spread) is shown in pink. The seasonal amplitude derived from the multi-model mean fields (red solid line), the multi-model mean of the seasonal amplitude from each model (red dashed line), and ozonesonde observations with a bin of 5 degrees (black diamonds) are also shown. From top to bottom, results are shown for 200 hPa, 500 hPa, and 800 hPa.

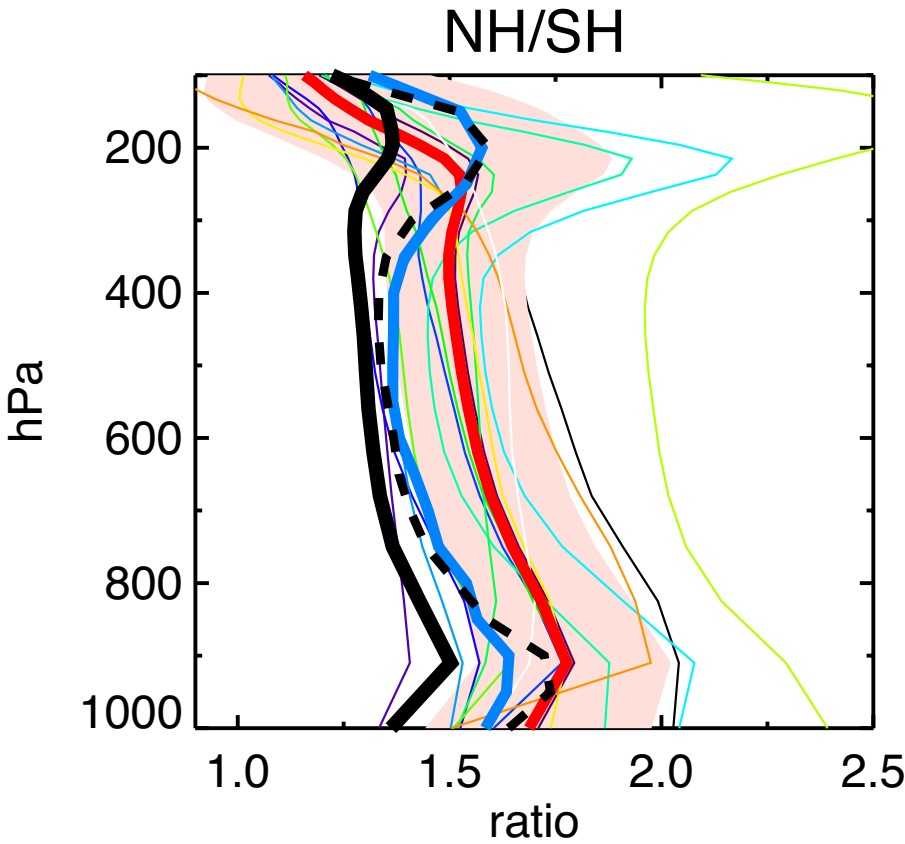

**Figure 7.** Vertical profile of inter-hemispheric gradient of annual mean ozone concentrations estimated from the reanalysis (black lines), ACCMIP ensemble mean (red solid line), ACCMIP models (thin colored lines), and ozonesonde observations (blue solid line). The reanalysis result is shown for the average over all model grid points (black solid line) and over the ozonesonde samplings (black dashed line). The $\pm 1\sigma$ deviation among the ACCMIP models is shown in pink.

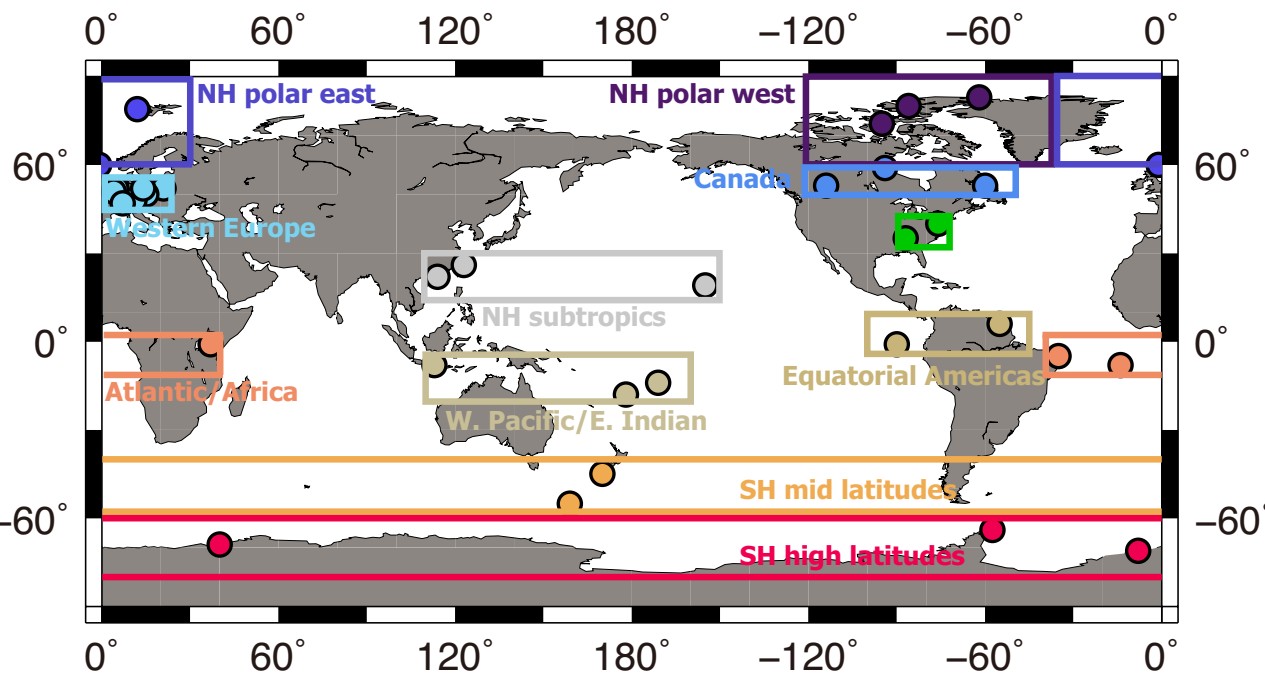

**Figure 8.** Regions and observation sites used in model evaluation. The 11 regions are defined following Tilmes et al. (2012). See also Table 3.

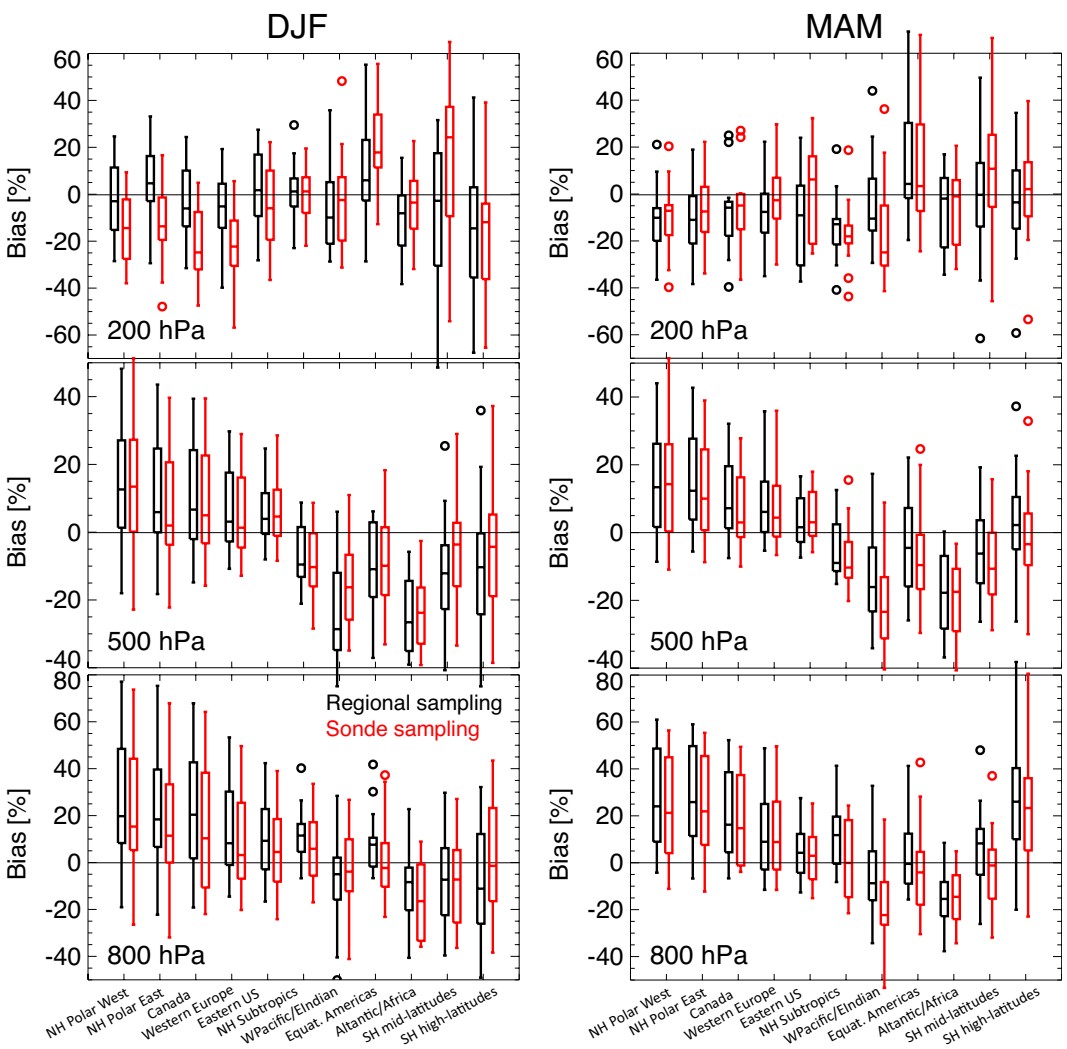

**Figure 9.** Box plots of relative model–reanalysis difference for seasonal mean concentration for DJF (left) and MAM (right) at 200 hPa (top), 500 hPa (middle), and 800 hPa (bottom). Results are shown for ACCMIP model simulations for 11 regions (c.f., Table 3 and Fig. 8). Black box shows model minus reanalysis difference for regional mean concentration (averaged over all model grid points); red box shows model minus reanalysis at the ozonesonde samplings.

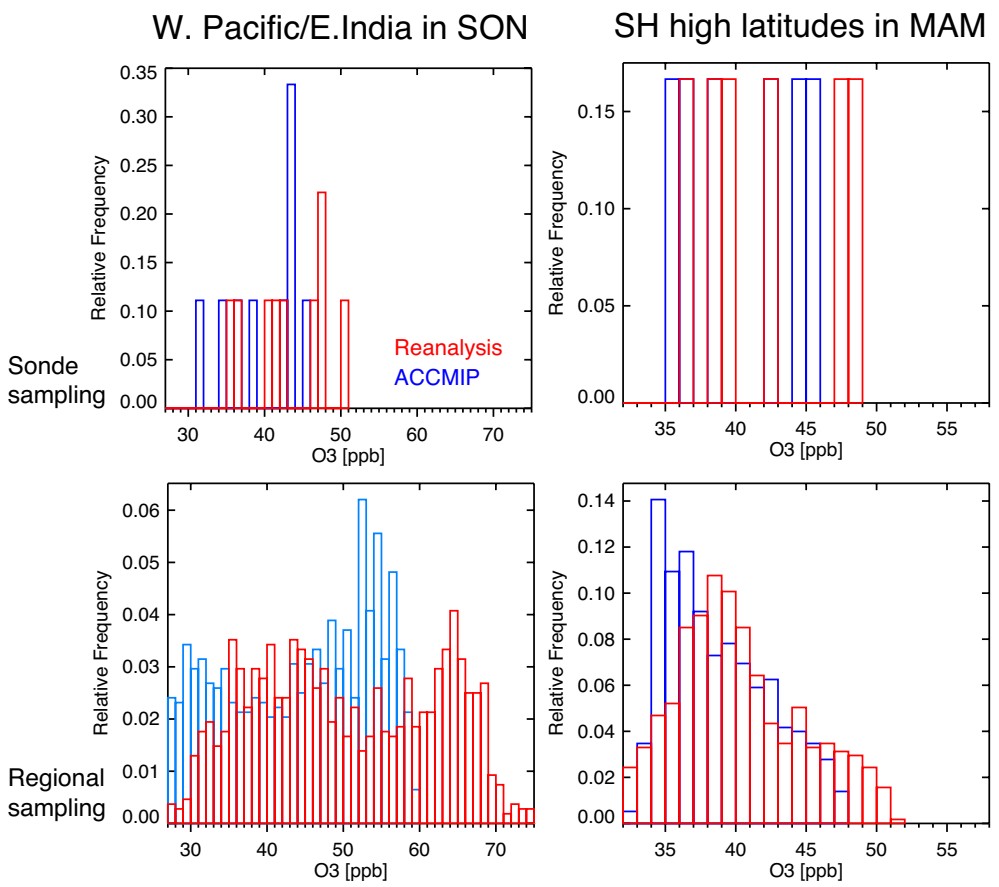

**Figure 10.** Probability distribution functions (PDFs) of ozone concentration obtained from the ACCMIP multi-model mean (blue) and the reanalysis (red) at 500 hPa for W. Pacific/E. India in SON (left) and for the SH high latitudes in MAM (right). The plots are shown for all model and reanalysis grid point (bottom) and for the ozonesonde sampling (top) within each defined region.

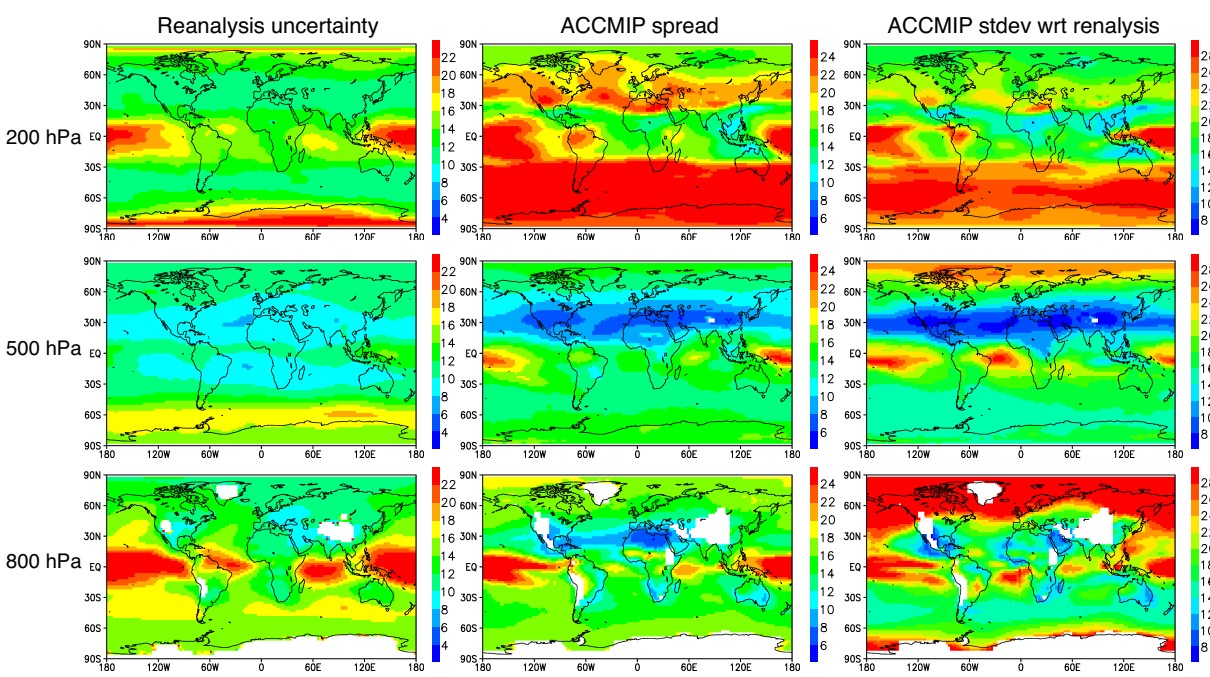

**Figure 11.** Global distributions of relative value (in %) of reanalysis uncertainty (left), standard deviation among the ACCMIP models (center), and ACCMIP model standard deviation with respect to the reanalysis for the annual mean concentration (right). From top to bottom, results are shown for global distributions at 200 hPa, 500 hPa, and 800 hPa.