# Peer review of "Evaluation of ACCMIP ozone simulations and ozonesonde sampling biases using a satellite-based multi-constituent chemical reanalysis"

_Atmospheric Chemistry and Physics, 2016_

## Referee Comment (RC1) · Anonymous Referee #2 · 20 Jan 2017

Review of Evaluation of ACCMIP ozone simulations using a mulch-constituent chemical analysis by Miyazaki and Bowman for ACP

In this work, the authors evaluate tropospheric ozone in the 2000 time-slice simulations performed by global chemistry-climate models in support of the Atmospheric Chemistry-Climate Model Intercomparison Project using a chemical reanalysis product that assimilates data from multiple satellites. The chemical reanalysis for 2005-2009 is first evaluated against global ozonesonde data and found to be in good agreement with measurements. In comparison to the reanalysis, the model ensemble mean is found to underestimate tropospheric ozone spatial and temporal variability. The paper is mostly well-written and is within the scope of the journal.

[Figure]

Evaluation against observation is a necessary step for building confidence in the global model simulation of tropospheric ozone. Limited observational constraints restrict our ability to thoroughly evaluate models. The use of chemical reanalysis discussed in this work is a promising method of model evaluation. However, a major aspect of the evaluation presented in this work need to be addressed before I can recommend publication. Reanalysis over four years (2005-2009) is used to evaluate time-slice simulations characteristic of the year 2000. The ACCMIP simulations were designed to eliminate interannual variability whereas the reanalysis product includes interannual variability. The use of 4-year product for evaluating climatological mean model simulations needs to be strongly justified.

Below are additional specific comments to help improve the manuscript.

Specific Comments:

Abstract: Please mention that a 4-year reanalysis data is used to compare ACCMIP time-slice simulations, and that the evaluation itself can be biased because of this inconsistency.

P1L17: Insert anthropogenic between "...important greenhouse.."

P2L1: There are several studies highlighting the use of CTMs/CCMs to assess the radiative impacts of tropospheric ozone prior to Bowman et al (2013). Please acknowledge those.

P2: 2nd and 3rd paragraphs discuss uncertainties in measurements for evaluating chemistry-climate models. I think they can be combined and modified for clarity.

P3L12: Please provide motivation for evaluating only tropospheric ozone and not its precursors (e.g., NO2, CO). Presumably biases in ozone are driven by biases in its precursors.

P4. How is photolysis calculated in the forecast model? Does the model represent methane - prescribed concentrations or emissions? Since the ACCMIP models used

different emissions inventory for ozone precursors compared to what is used in this reanalysis product, how would the comparison be affected by this difference?

P4L13: What is the convection scheme of MIROC-AGCM? Please describe it in a sentence.

P5L7: What is DOMINO data? What does the acronym stand for?

P5L21: According to Lamarque et al (2013), models were run for several years (up to 10 years) for each time slice. How are the model results for 2000 time-slice handled for comparison with the reanalysis?

P5L29: Young et al. (2013) note that although the models used the same anthropogenic and biomass burning emissions, model-to-model diversity in the implemented chemical scheme resulted in differences in precursor (especially VOCs) emissions across the models. So, the statement "same emissions were used in all the models" is not accurate. Please modify.

P6L7: Please provide details on how the ACCMIP model monthly ozone concentrations were interpolated to 2 hour temporal resolution? What diagnostics from the reanalysis were used to compare with observations - monthly or hourly ozone? Some clarification is needed here.

P6L29: Can you please elaborate on the model setting that caused this significant degradation of the representation of ozone in the UTLS?

P7L3: The terminology to refer to model output is somewhat confusing making it difficult to keep track of observations versus model output. Suggest referring to output from the forecast model and ACCMIP as "forecast model output" and "ACCMIP model output", respectively, and observations as "data".

P7L22-24: The simulation of the "wave-1" pattern by ACCMIP models has already been highlighted by Young et al. (2013). This reference needs to be cited.

P7L32: It is not clear what "common reported" means here. Please clarify.

P8L10: Which region is "In this region…" referring to?

P8L14-15: These results are consistent with those discussed in Young et al. (2013).

P8L24: The differences could also be associated with the way biomass burning emissions are handled across models - whether they are emitted at the surface layer or distributed vertically in the model, as a larger role for biomass burning in the tropical mid to upper troposphere has been suggested recently (Anderson et al., 2016).

P8L33-34: Why are results from a specific model highlighted? Is it because model 8 is driving the large model diversity at 200hPa? Please provide a figure to support this statement if this is indeed the case.

P9L3: It took me a while to understand the meaning of "reanalysis concentrations from the ozonesonde sampling". Please rephrase this to indicate that reanalysis output is sampled at ozonesonde sites instead of averaging the reanalysis at all grid cells.

P929-31: Here again the results are consistent with Young et al. (2013) - see their Figure 4 and its discussion.

P10L23-25: Please clarify what is meant by "radiative heating distribution in chemistry-climate simulations are largely uncertain…" Are you referring to the radiative heating due to tropospheric ozone? Please elaborate on how the O3 NH/SH ratio provides information on the radiative heating distribution.

P11L17: replace "completion" with compilation.

P11L19: By coarse resolution, do you mean the coarse horizontal resolution?

P11L29: Did you mean - "...less variabilities in the SH than in the NH"?

P12L9-11: Please quantify large in "Large negative model biases…" and larger in "...errors are larger than those…"

Section 5.1 Please elaborate on how the comparisons discussed here may be influenced by inconsistency in the time period of the reanalysis and the ACCMIP simulations. The reanalysis output is for the 2005-2009 whereas the ACCMIP simulations are representative of 2000. Additionally, the precursor emissions used in ACCMIP simulations were decadal means and not specific to the year of simulation. For example, year 2000 biomass burning emissions were calculated as average over 1997–2006 (Lamarque et al., 2010), so they encompass the high emissions over Southeast Asia in 1998, an El Nino year. One would expect that biases due to sampling in time would occur similar to the biases due to sampling in space.How significant are the biases due to spatial sampling errors (highlighted here) compared to temporal sampling errors. The issue of temporal and spatial sampling was recently highlighted by Lin et al. (2016) in the context of tropospheric ozone trends.

P14L24: Replace biogenetic with biogenic.

P15L6: I am not sure if the 2005-2009 can be considered a "long-record" Are the authors referring to the possibility of a long-record reanalysis sometime in the future when observations have accumulated in time.

P15L7-9: From Lamarque et al. (2013): "This averaging was designed to reduce the effect of interannual variability and therefore provide optimal conditions from which average composition changes and associated forcings can be more readily computed." The ACCMIP simulations were designed to remove interannual variability, therefore, it is unjustified to state that "the influence of ENSO was not well-simulated in ACCMIP..."

P33 Figure 5. Please provide statistics such as mean bias and correlation for the comparisons here.

References:

Anderson et al., A pervasive role for biomass burning in tropical high ozone/low water structures, Nature Communications 7, Article number: 10267 (2016)

[Figure]

doi:10.1038/ncomms10267.

Lin, M., L. W. Horowitz, O. R. Cooper, D. Tarasick, S. Conley, L. T. Iraci, B. Johnson, T. Leblanc, I. Petropavlovskikh, and E. L. Yates (2015), Revisiting the evidence of increasing springtime ozone mixing ratios in the free troposphere over western North America, Geophys. Res. Lett., 42, 8719–8728, doi:10.1002/2015GL065311.

————————————————————

---

## Referee Comment (RC2) · Anonymous Referee #1 · 27 Jan 2017

Evaluation of ACCMIP ozone simulations using a multi-constituent chemical reanalysis by Kazuyuki Miyazaki and Kevin Bowman

Overview:

The authors use a new chemical reanalysis based on multiple satellite observations to evaluate ACCMIP ozone simulations for the period 2005-2009. This evaluation is juxtaposed with an evaluation using ozone sondes. Thereby, the authors quantify the sampling biases of the ozone sonde observations and its impact on the evaluation results. The evaluation with the ozone re-analysis, which was shown to be in good agreement with the ozone sonde observations, give a more comprehensive picture of the model deficiencies than ozone sondes.

[Figure]

General remarks:

Chemical reanalyses are newly emerging data sets. The authors present an interesting and convincing application of these data set for the evaluation of atmospheric chemistry models. To make this point they show that the evaluation with ozone sondes can suffer from biases because of the spatial coverage and representativeness as well as the specifics of the temporal sampling of the ozone sonde network. The work is timely and of high interest to the scientific community.

The paper would benefit from a more stringent focus on the ozone sonde sampling biases and its impact on the evaluation because this is the actual novelty aspect of the paper. The ACCMIP models have been evaluated. So only the differences of the new evaluation approach with previous work is of interest. I would recommend to add to the title "- focus on ozone sonde sampling biases" or similar. The sampling biases should be mentioned and discussed in abstract and introduction more clearly.

To get a better understanding of the sampling biases, i.e. the difference in the mean over area averages using all grid points at regular intervals or only the stations locations at the time of the observations, it is recommended to show the sampling biases not only for the differences between ACCMIP models and the reanalysis but also for the Re-analysis and the model runs, including the control run itself. It would be interesting to see to what extent they differ as the reanalysis may also be effected by the "sampling biases" of the assimilated observations. A strong sampling biases for model result will help to convince modellers to use reanalysis data for model evaluation.

The authors should aim to provide a better understanding of the reasons of the sampling biases. Do they come more from spatial heterogeneity or the variable temporal sampling. The latter can be estimated by comparing re-analysis means at a 2 hourly resolution or only at the ozone sonde observing times.

For the sake of consistency the quantification of the sampling biases should be carried out for one set of latitude bands in the same way as in the more regional analysis

presented in section 5.

Section 5 "Impact of Sampling on model evaluation" discusses the regional biases and the general problem in a lot of detail but sections 4.2 and 4.3 discuss already the sampling biases for the latitude bands. I recommend moving the introduction of the sampling biases to an earlier section (2).

The discussions section, in particular 6.3, does not discuss the direct results of the paper but gives an outlook on other potential aspect of the usefulness of the evaluation with chemical re-analyses. However, the positive impact on species not directly assimilated has not be demonstrated in the paper. Also, the four year comparison is not long enough to infer trends and longer re-analysis of atmospheric composition are likely to suffer from temporal artefacts because of the changing observational system. I would therefore not discuss in detail these aspects in the paper as there is not enough evidence given to support them.

The used ozone sondes observations need to be clearer identified and their sampling discussed. A table of the used ozone sondes, their sampling frequency and outage in the period and mean should be summarised in a table not only for the regional areas but also for the latitude bands. It should be made clear which stations are used for global/hemispheric stratification in Figures 1 to 7 and the more regional stratification Figure 9-10.

Specific remarks,

PL1: 5 Please "the" before instrument names

P1L5: Please add a sentence on the advantages of using a 3D re-analysis rather than ozone sondes for the model evaluation.

P1L6: Please ad here or at L 12 the problem of the ozone sampling biases

P1L12: Please state more clearly the differences in the evaluation results when using the re-analysis as complete field and on the the ozone sonde observation locations

and times only.

P1L24: better "transport"

P2L8: Please add some references for these evaluation studies

P2L13: there is a "First" and a "Third" (L19) but I did not find a "Second"

P2L18: The sentence starting with "However, . . ." is a strong motivation for the paper. Please elaborate and also mention that the climatologies do no capture the temporal variability of the observed ozone.

P2L30: Please consider citing overview papers such as Bocquet et al. (ACP 2015) or Sandu et al. (Atmosphere, 2011)

P4L4: Please comment how this is related to resolution of the evaluated ACCMIP models.

P4 L22: Please explain how the ensemble is constructed, i.e. what parameters are varied to get a different ensemble members in the EnKF. This information is important because you later use the ensemble spread partially as indicator of the analysis error.

P4 L23: "satellite retrieval operator"? This implies radiances i.e. Level 1 were assimilated, which is perhaps not the case. Please clarify.

P5L4: Please mention if it could be shown that the modulation of the lifetimes was an improvement.

P5 L14: Please provide a table with the assimilated retrievals and additional information such as assimilated height range, temporal data coverage and an indication of observation errors statistics.

P5L21: Please elaborate on the period and the meteorological input for this time-slice setup. It is important to know what sort of realism can be expected from the simulation if they are compared against observations.

P6L9: Please clarify which station were used for the comparison. The ones listed in table 4 ? If so mention it here. Provide information about station numbers and individual temporal coverage as this may vary greatly and contribute to the ozone sonde sampling bias.

P6 L17-23: This description of the model changes may better put in the model description section.

P6L30: Please clarify what the differences in the assimilated observations are between this data set and the previous one.

P7L10: Please clarify what temporal averaging the temporal correlation is based on (i.e. monthly means, annual means, instantaneous values etc.). It is good practise to de-seasonalize the time series to get a more meaningful information about the temporal correlation. On the other hand, 5 years might be too short to obtain a robust information about seasonality and year-to-year variability.

The numbers in figure 1 indicate a very good reduction in biases but far less so for variability measures. (The reduction in RMSE seems dominated by the bias component and temporal correlation is less improved). This seems to contradict the theoretical basis of data assimilation, which is meant to reduce the error variance assuming bias free model. A further discussion would be helpful.

P7L14: As you also show the ozone sondes in the sections on seasonal variation and hemispheric gradient, it seems odd not to show the ozone sondes observations in Figure 3. Please add a further panel with colour dots at the station location.

P7L18: It is confusing that you choose a different latitude bands for table 1 and table 2. Please use the same selection of latitude bands through the paper.

P7L18: Please, clarify how spatial r was calculated (only using the 5 year mean, based on lat-long grid points or area-weighted grid-points etc. Consider filtering small scale noise by averaging over areas corresponding to the resolution of the reanalysis.

The spatial correlation coefficient presented here seems less suited to express agreement in spatial patterns, which would be meaningful for the understanding of the model performance. Spatial r might be too much effected by the underlying spatial variability of the actual fields, thereby penalizing fields with greater more random variability i.e. standard deviation.

P7L19: The lower spatial correlation coefficient at p=500 hPa in NH could simply be caused by a different transport patterns (winds) and larger heterogeneity than in SH. Good correlation at the surface could be simply because a good match of emission patterns. High correlation at 200hPa in extra-tropics could mean that the transition in to the stratosphere agreed reasonably well. So are the different spatial r really helpful to distinguish model performance?

P8L4: Please clarify again how the statistical variables shown in the Taylor diagram were computed. Given my scepticism about the meaning of the spatial correlation, I would consider omitting Figure 4 and shortening the discussion.

P9L6: Please clarify how the seasonal amplitude was calculated. How was made sure that "noise", i.e. unstructured variability, was not attributed to the seasonal amplitude.

P9L12: In section5 you discuss the sampling bias with respect to the regional areas. You should also discuss the sampling biases w.r.t to the selected latitude bands. This is needed because you also discuss model performance for the latitude bands. As mentioned in the general remarks, please also indicate the difference between the model results sampled at ozone locations and observation times and the area averages.

P9L13: Please confirm that the average is area weighted and not based simply on lat-long grid boxes, which decrease in size towards the poles.

P10L8: Please clarify how you exactly calculate the hemispheric gradient both for the gridded fields and for the ozone sonde observations.

P10L15: add missing "At" before "Around "

[Figure]

P10L18: How do these value compare to values from the literature?

P10L26: As you already discuss sampling biases it a bit inconsistent to put the section at this place. This very good introduction to sampling biases (p10L27 – p11L15) should come earlier in the paper, i.e. in the part when you discuss the methods (section2)

P10L15: The sampling biases depends on the averaging area and the selection of ozone sondes. The sampling biases estimated by using your re-analysis should be presented for the Tilmes regions as well as for the latitude bands (choose one set only) in and uniform way. As model results are often evaluated for the latitude bands , this information would be very interesting for the scientific community.

P10L24 Please add also the stations used for the latitude bands averages in table 3.

P12L1: I think there is would be very good to compute the sampling biases also directly for the re-analysis i.e. the difference between the re-analysis sampled as ozone and as area-time averages. This information would be in my opinion of more general meaning than the values for the ACCMIP error.

P12L1: Why does table 5 show the median whereas otherwise only the ensemble mean is discussed or shown. (Using only the median would be perhaps a better option overall)

P13L8: The ozone network in SH high-latitudes is actually quite high because of the need to monitor the ozone hole. The launch frequency varies for some stations a lot because more sondes are launched during the ozone hole season.

P13L24: Please add also the sampling biases for the latitude bands.

P14 L14: Please clarify how much of the analysis ensemble spread depends on sometimes arbitrary choices to cause spread between the ensemble members.

P14L15: Please clarify to what extent the analysis uncertainty is controlled by the uncertainty of the assimilated observations.

P14L21: How does the ensemble spread relate to the spread of the ensemble in the EnKF. Could the ACCMIP ensemble spread be used to verify the EnKF ensemble spread ?

P15L3: Please see my general comment on this chapter. Improvement on species not directly assimilated needs to be demonstrated. Long-term reanalysis could suffer from artificial jumps because of the change in the observing system (for example degradation of TES after 2010).

P15L29: I don't understand this conclusion at all. Re-analysis are only valid for present day conditions when observations are available. They cannot be used to for pre-industrial estimates nor the differences with today's values.

P16L4: Please mention that you (only) consider ozone sondes as reference in this paper.

P16L6: Please mention the advantage of using a re-analysis, i.e. a gridded field. Please mention that the biases of the re-analysis against ozone sondes are small.

P16L17: Please add a statement if these finding are consistent with other evaluation studies, i.e. the Young et al. paper.

P16L20-30: Please give some numbers for the sampling biases. Also include the sampling bias w.r.t to latitude bands. p17L4: Please add a statement that it will be a challenge to combine all these observations in a consistent way in a more long term re-analysis.

---

## Author Comment (AC1) · 5 Apr 2017

**Authors' comments in reply to the anonymous referee for "Evaluation of ACCMIP ozone simulations using a multi-constituent chemical reanalysis" by K. Miyazaki and K. Bowman**

We want to thank the referee for the helpful comments. We have revised the manuscript according to the comments, and hope that the revised version is now suitable for publication. Below are the referee comments in italics, with our replies in normal font.

*Reply to Referee #1*

*General remarks:*

*The paper would benefit from a more stringent focus on the ozone sonde sampling biases and its impact on the evaluation because this is the actual novelty aspect of the paper. The ACCMIP models have been evaluated. So only the differences of the new evaluation approach with previous work is of interest. I would recommend to add to the title "- focus on ozone sonde sampling biases" or similar. The sampling biases should be mentioned and discussed in abstract and introduction more clearly.*

The title has been revised as follows:
"Evaluation of ACCMIP ozone simulations and ozonesonde sampling biases using a satellite-based multi-constituent chemical reanalysis."

The abstract and introduction have been modified to mention the sampling biases.

*To get a better understanding of the sampling biases, i.e. the difference in the mean over area averages using all grid points at regular intervals or only the stations locations at the time of the observations, it is recommended to show the sampling biases not only for the differences between ACCMIP models and the reanalysis but also for the Re-analysis and the model runs, including the control run itself. It would be interesting to see to what extent they differ as the reanalysis may also be effected by the "sampling biases" of the assimilated observations. A strong sampling biases for model result will help to convince modellers to use reanalysis data for model evaluation.*

Table 9 has been added to discuss the impacts of the sampling biases in the reanalysis and control run comparisons. These results are discussed in Section 5.1 as follows:
"Further, ozonesonde sampling bias is evaluated for the control run and reanalysis comparisons. As summarized in Table 9, at 500 hPa, there are large differences (> 30 %) between the two evaluations in

many regions, especially in the NH mid latitude regions in winter and in the tropics throughout the year, as also found in the ACCMIP models and reanalysis comparisons (Table 8). The analysis increments introduced by data assimilation vary with space and time, reflecting the changes in coverage and uncertainty of assimilated measurements as well as in model errors. Nevertheless, observational information was propagated globally and integrated with time through forecast steps during the data assimilation cycles. This is true for ozone because of its relatively long lifetime in the free troposphere. Therefore, the spatial distribution is well constrained by data assimilation, and we do not expect large variations in the reanalysis quality within each analysis region."

*The authors should aim to provide a better understanding of the reasons of the sampling biases. Do they come more from spatial heterogeneity or the variable temporal sampling. The latter can be estimated by comparing re-analysis means at a 2 hourly resolution or only at the ozone sonde observing times.*
*For the sake of consistency the quantification of the sampling biases should be carried out for one set of latitude bands in the same way as in the more regional analysis presented in section 5.*

Table 8 has been revised to discuss the influences of temporal and spatial sampling errors separately. The following sentences have been added in Section 5.1:
"Our analysis using monthly reanalysis fields sampled at the ozonesonde locations (brackets in Table 8) suggests a greater impact of the spatial sampling bias than the temporal sampling bias for the NH polar east in DJF."
"At 500 hPa over Canada, the relative importance of the spatial and temporal sampling biases varies with season: the spatial (temporal) sampling bias is dominant in DJF (JJA), whereas both of them are important in MAM."
"Over the Western Pacific and East Indian Ocean, the sampling bias is not reduced by using monthly mean reanalysis fields (sampled at the ozonesonde locations) in DJF and JJA. This suggests that ozone varies with time and space in a complex manner, and a dense (in both space and time) network would be required to capture the regional and seasonally representative model biases in this region."
"The temporal sampling bias mostly dominates the difference in the SH high latitudes in MAM and JJA, whereas the spatial sampling bias is also important in the SH mid latitudes in DJF and MAM."

Table 8 has been modified to describe the sampling biases for four latitude bands. The following sentences have been added in Section 5.1:
"Table 8 also shows the model evaluation results for four latitudinal bands at 500 hPa. The observations used are shown in bold in Table 2. The differences between the two evaluations are small in the NH extratropics (30-90N) in all seasons, because of the relatively large number of observations. There are

large differences in the tropics of both hemispheres: the ozonesonde network reveals a large negative sampling bias in the model evaluation in the NH tropics (Eq-30N) in SON (-9 % in the complete sampling and -16 % in the ozonesonde sampling) and in the SH tropics (30S-Eq) in MAM (-14 % and -21 %) and a large positive sampling bias in the NH tropics in JJA (-7 % and -3 %). Large sampling biases (> 60 %) also exist in the SH extratropics (90-30S) in DJF and MAM due to the sparse ozonesonde network."

*Section 5 "Impact of Sampling on model evaluation" discusses the regional biases and the general problem in a lot of detail but sections 4.2 and 4.3 discuss already the sampling biases for the latitude bands. I recommend moving the introduction of the sampling biases to an earlier section (2).*

The introduction and methodology of the sampling biases have been moved to Section 2.4 (Section title: Ozonesonde sampling bias estimation).

*The discussions section, in particular 6.3, does not discuss the direct results of the paper but gives an outlook on other potential aspect of the usefulness of the evaluation with chemical re-analyses. However, the positive impact on species not directly assimilated has not be demonstrated in the paper. Also, the four year comparison is not long enough to infer trends and longer re-analysis of atmospheric composition are likely to suffer from temporal artefacts because of the changing observational system. I would therefore not discuss in detail these aspects in the paper as there is not enough evidence given to support them.*

Because this is the first study to use chemical reanalysis for model evaluation, it is worthwhile discussing its possible application in future studies. The positive impacts on non-assimilated species have been discussed in our previous studies, and this is described in the revised manuscript as follows:
"Miyazaki et al. (2012b, 2015) demonstrated that the multiple-species assimilation results in a strong influence on both assimilated and non-assimilated species."

The limitation of the evaluations using the five-year (2005-2009) reanalysis is discussed as follows in Section 5.1 of the revised manuscript:
"The five-year reanalysis (2005-2009) may cause biases in the estimated model errors in the evaluation of the 2000 decade ACCMIP simulations that used decadal-averaged SST boundary conditions and biomass-burning emissions averaged over 1997--2006 (Lamarque et al., 2010). It may neglect the influences of interannual and decadal changes in both anthropogenic and biomass emissions and meteorology. Longer-term reanalysis and time-consistent validation are required to obtain more robust

error estimations.

To discuss remaining issues with a longer-term reanalysis, the following sentences have been added in Section 6.3:

"However, any discontinuities in the availability and coverage of the assimilated measurement will affect the quality of the reanalysis and estimated interannual variability, which limit the usability of a long term reanalysis for model evaluation, as discussed in Miyazaki et al (2015) for chemical reanalyses and in Thorne and Vose (2010) for climate reanalyses. This also requires a bias-correction procedure for each assimilated measurement, in order to improve the reanalysis quality (Inness et al, 2013)."

*The used ozone sondes observations need to be clearer identified and their sampling discussed. A table of the used ozone sondes, their sampling frequency and outage in the period and mean should be summarised in a table not only for the regional areas but also for the latitude bands. It should be made clear which stations are used for global/hemispheric stratification in Figures 1 to 7 and the more regional stratification Figure 9-10.*

Table 2 has been added.

*Specific remarks,*

*PL1: 5 Please "the" before instrument names*

Added.

*P1L5: Please add a sentence on the advantages of using a 3D re-analysis rather than ozone sondes for the model evaluation.*

The following sentence has been added:

"The reanalysis provides comprehensive information on the weakness of the models, whereas we consider that the spatial and temporal coverage of individual measurements, such as ozonesonde measurements, is insufficient to capture the temporally and spatially representative model bias."

*P1L6: Please ad here or at L 12 the problem of the ozone sampling biases*
*P1L12: Please state more clearly the differences in the evaluation results when using the re-analysis as complete field and on the the ozone sonde observation locations and times only.*

The following sentence has been added:

"The ozonesonde sampling bias in the evaluated model bias for the seasonal mean concentration is 40-50 % over the Western Pacific and East India and reaches 110 % over the equatorial Americas in the middle troposphere."

*P1L24: better "transport"*

Replaced.

*P2L8: Please add some references for these evaluation studies*

Added.

*P2L13: there is a "First" and a "Third" (L19) but I did not find a "Second"*

Corrected.

*P2L18: The sentence starting with "However, . . ." is a strong motivation for the paper. Please elaborate and also mention that the climatologies do no capture the temporal variability of the observed ozone.*

The sentences have been rewritten as follows:

"However, the climatological data does not provide information on the temporal variability of the observed ozone. In addition, the current ozonesonde network does not cover the entire globe and is not homogeneously distributed between the hemispheres, ocean and land, and urban and rural areas, and its sampling interval is typically a week or longer. Model errors are also expected to vary greatly in time and space at various scales."

*P2L30: Please consider citing overview papers such as Bocquet et al. (ACP 2015) or Sandu et al. (Atmosphere, 2011)*

Added.

*P4L4: Please comment how this is related to resolution of the evaluated ACCMIP models.*

The following sentence has been added:

"The horizontal model resolution is comparable to the resolution of ACCMIP models (ranging from 1.24° to 5°)."

*P4 L22: Please explain how the ensemble is constructed, i.e. what parameters are varied to get a different ensemble members in the EnKF. This information is important because you later use the ensemble spread partially as indicator of the analysis error.*

The following sentence has been added:

"The ensemble perturbations were introduced to all the state vector variables as described below."

*P4 L23: "satellite retrieval operator"? This implies radiances i.e. Level 1 were assimilated, which is perhaps not the case. Please clarify.*

The sentence has been rewritten as:

"and an operator that converts the model fields into retrieval space"

*P5L4: Please mention if it could be shown that the modulation of the lifetimes was an improvement.*

The following sentence has been added:

"Miyazaki et al. (2015) demonstrated that the Northern/Southern Hemisphere OH ratio became closer to an observational estimate of Patra et al (2014) due to the multiple-species assimilation."

*P5 L14: Please provide a table with the assimilated retrievals and additional information such as assimilated height range, temporal data coverage and an indication of observation errors statistics.*

Table 1 has been added.

*P5L21: Please elaborate on the period and the meteorological input for this time-slice setup. It is important to know what sort of realism can be expected from the simulation if they are compared against observations.*

The following sentences have been added:

"The number of years that the ACCMIP models simulated for the 2000 decadal simulation mostly varied between 4 and 12 years for each model. Each model simulation was averaged over the simulated years."

"Meteorological fields were obtained from analyses in CICERO-OsloCTM2 and from climate model fields in MOCAGE. UM-CAM and STOC-HadAM3 simulated meteorological and chemical fields, but chemistry did not affect climate. In all other models, simulated chemical fields were used in the radiation calculations and hence provide a forcing effect on the general circulation of the atmosphere. Lamarque et al. (2013) indicated that most models overestimate global annual precipitation and have a cold bias in the lower troposphere."

*P6L9: Please clarify which station were used for the comparison. The ones listed in table 4 ? If so mention it here. Provide information about station numbers and individual temporal coverage as this may vary greatly and contribute to the ozone sonde sampling bias.*

The sentences have been rewritten as:

"All available data from the WOUDC database are used for the evaluation of reanalysis data (Section 3), as listed in Table 2. For the evaluation of ACCMIP models and ozonesonde sampling biases (Section 4 and 5), we use the ozonesonde sampling based on the compilation by Tilmes et al. (2012), which is shown in bold in Table 2. Because there is no observation after 2003 in Scoresbysund, this location has been removed from the compilation in this study."

Table 2 has been added to summarize the ozonesonde measurements.

*P6 L17-23: This description of the model changes may better put in the model description section.*

Moved.

*P6L30: Please clarify what the differences in the assimilated observations are between this data set and the previous one.*

The following sentence has been added:

"MLS retrievals have been updated from v3.3 in Miyazaki et al. (2015) to v4.2 in this study."

*P7L10: Please clarify what temporal averaging the temporal correlation is based on (i.e. monthly means, annual means, instantaneous values etc.). It is good practise to de-seasonalize the time series to get a more meaningful information about the temporal correlation. On the other hand, 5 years might be too short to obtain a robust information about seasonality and year-to-year variability.*

The sentence has been rewritten as follow:

"The tropospheric concentrations show distinct seasonal and year-to-year variations, for which the temporal correlation based on the monthly and regional mean concentrations is increased by the data assimilation globally, except at high latitudes in the lower troposphere."

Because the seasonal pattern varies with year especially in the tropics, we did not apply de-seasonalization.

The limitations of the five-year reanalysis data are discussed in the revised manuscript as follow:
"The reanalysis can be extended to a longer-term validation that will provide more information on seasonality and year-to-year variability."

*The numbers in figure 1 indicate a very good reduction in biases but far less so for variability measures. (The reduction in RMSE seems dominated by the bias component and temporal correlation is less improved). This seems to contradict the theoretical basis of data assimilation, which is meant to reduce the error variance assuming bias free model. A further discussion would be helpful.*

We confirmed that both the bias and RMSE are largely reduced compared with assimilated measurements (e.g., TES) due to data assimilation, as demonstrated in our previous study (Miyazaki et al., 2015). In the comparison against independent ozonesonde measurements in this study, spatial gaps between the model/analysis and observations (i.e., representativeness error) result in large RMSE even after data assimilation.

The relevant sentence has been rewritten as follows:
"Root-Mean-Square-Errors (RMSEs) are also reduced above the middle troposphere, although the reduction rate is relatively small compared to the bias, probably due to representativeness errors between the ozonesonde measurements and data assimilation analysis."

*P7L14: As you also show the ozone sondes in the sections on seasonal variation and hemispheric gradient, it seems odd not to show the ozone sondes observations in Figure 3. Please add a further panel with colour dots at the station location.*

Added.

*P7L18: It is confusing that you choose a different latitude bands for table 1 and table 2. Please use the*

*same selection of latitude bands through the paper.*

Corrected.

*P7L18: Please, clarify how spatial r was calculated (only using the 5 year mean, based on lat-long grid points or area-weighted grid-points etc. Consider filtering small scale noise by averaging over areas corresponding to the resolution of the reanalysis.*

The sentence has been rewritten as:

"As summarized in Table 4, the global spatial distributions are similar between the five-year mean reanalysis field and the ensemble mean when estimated at 2°×2.5° spatial resolution, with a spatial correlation (r) greater than...".

In addition, the following sentence has been added in Section 2.2:

"Both the ACCMIP models and chemical reanalysis are interpolated to at 2°×2.5° spatial resolution and 67 levels, following Bowman et al. (2013), and then compared each other. Spatial correlations are computed with consideration of weighting for the latitude."

*The spatial correlation coefficient presented here seems less suited to express agreement in spatial patterns, which would be meaningful for the understanding of the model performance. Spatial r might be too much effected by the underlying spatial variability of the actual fields, thereby penalizing fields with greater more random variability i.e. standard deviation.*

Because all model and reanalysis fields were interpolated into the same spatial resolution (2°×2.5°) before the comparisons, the estimated spatial correlation can provide information on the model performance on the spatial pattern at that spatial scale. Although more thorough evaluations would be required for more careful discussions of the spatial pattern, the present evaluation method has been widely used and is valid.

*P7L19: The lower spatial correlation coefficient at p=500 hPa in NH could simply be caused by a different transport patterns (winds) and larger heterogeneity than in SH. Good correlation at the surface could be simply because a good match of emission patterns. High correlation at 200hPa in extra-tropics could mean that the transition in to the stratosphere agreed reasonably well. So are the different spatial r really helpful to distinguish model performance?*

The different reasons for each region raised by the reviewer are discussed in the manuscript. Because the spatial correlation varies significantly among the models as discussed in Section 4.1, it is a useful diagnostic of model performance. Please also see our reply above.

*P8L4: Please clarify again how the statistical variables shown in the Taylor diagram were computed. Given my scepticism about the meaning of the spatial correlation, I would consider omitting Figure 4 and shortening the discussion.*

The following sentence has been added in Section 2.2:
"Both the ACCMIP models and chemical reanalysis are interpolated to at 2º×2.5º spatial resolution and 67 levels, following Bowman et al. (2013), and then compared each other. Spatial correlations are computed with consideration of weighting for the latitude."

We think the Taylor diagram plots are useful to measure the general performance of each model and are widely used in the climate model evaluation. Please also see our reply on the spatial correlation estimates above.

*P9L6: Please clarify how the seasonal amplitude was calculated. How was made sure that "noise", i.e. unstructured variability, was not attributed to the seasonal amplitude.*

The seasonal amplitude was estimated from the difference between maximum and minimum monthly mean concentrations, which could reflect noise in the seasonal variation. This is described in the revised manuscript.

*P9L12: In section5 you discuss the sampling bias with respect to the regional areas. You should also discuss the sampling biases w.r.t to the selected latitude bands. This is needed because you also discuss model performance for the latitude bands. As mentioned in the general remarks, please also indicate the difference between the model results sampled at ozone locations and observation times and the area averages.*

Please see our reply above.

*P9L13: Please confirm that the average is area weighted and not based simply on lat-long grid boxes, which decrease in size towards the poles.*

The average is area weighted, as described in the revised manuscript.

*P10L8: Please clarify how you exactly calculate the hemispheric gradient both for the gridded fields and for the ozone sonde observations.*

The following sentence has been added:
"For the estimation of the gradient using the ozonesonde observations, we made a gridded dataset from the ozonesonde observations based on the completion by Tilmes et al (2012) at $2^{\circ} \times 2.5^{\circ}$ spatial resolution, and then calculated area-weighted hemispheric mean concentrations using the gridded data."

*P10L15: add missing "At" before "Around "*

Added.

*P10L18: How do these value compare to values from the literature?*

To the best of our knowledge, there is no literature that shows an inter-hemispheric ozone gradient for different altitudes of the troposphere.

*P10L26: As you already discuss sampling biases it a bit inconsistent to put the section at this place. This very good introduction to sampling biases (p10L27 – p11L15) should come earlier in the paper, i.e. in the part when you discuss the methods (section2)*

Moved to Section 2.4.

*P10L15: The sampling biases depends on the averaging area and the selection of ozone sondes. The sampling biases estimated by using your re-analysis should be presented for the Tilmes regions as well as for the latitude bands (choose one set only) in and uniform way. As model results are often evaluated for the latitude bands, this information would be very interesting for the scientific community.*

Table 8 has been modified to include the results for four latitudinal bands.

*P10L24 Please add also the stations used for the latitude bands averages in table 3.*

This is mentioned in Section 2.3 and Table 2 in the revised manuscript.

*P12L1: I think there is would be very good to compute the sampling biases also directly for the re-analysis i.e. the difference between the re-analysis sampled as ozone and as area-time averages. This information would be in my opinion of more general meaning than the values for the ACCMIP error.*

Table 7 has been added to discuss the sampling bias for the reanalysis fields. The following discussion, regarding this table, has been added in Section 5:

"Table 7 demonstrates the regional and seasonal mean differences of the reanalysis concentrations between the complete sampling and the ozonesonde sampling. The ozonesonde sampling results have higher concentrations (by about 3 %) in the two NH polar regions for most cases, whereas the difference is smaller in NH polar west than in NH polar east. Among the NH mid-latitude regions, a large difference (about 14 %) exists between the two cases over the eastern United States in June-August (JJA), where the comparison using monthly reanalysis fields sampled at the ozonesonde locations (brackets in Table 7) suggests that the sampling bias is dominated by temporal variations. The tropical and subtropical regions exhibit large sampling biases, 4-12.3 % over the NH subtropics, -3.2-5.0 % over the Western Pacific and East Indian Ocean, 0--7.8 ¥% over the equatorial Americas, and -3.8-7.5 % over the Atlantic Ocean and Africa. In most of the tropical and subtropics regions, both the spatial and temporal sampling biases are important, because of large spatial and temporal variability of ozone and the sparse observation network. For the global tropics, the sampling bias reaches 13 % in the NH (Eq-30N) and 8 % in the SH (30S-Eq). Thus, the ozonesonde network has a major limitation when it comes to capturing ozone concentrations that are representative of seasonal and regional means for the entire tropical region. The sampling bias may not be negligible even in the SH (0.3-3.9 % in the SH mid-latitudes and 0.8-4.2 % in the SH high latitudes), and it is large (up to 13 %) when estimations are done for a large area (90-30S). The large sampling bias in 90-30S is primarily attributed to spatial variability. The impact of the sampling bias on the model evaluation is discussed in the following section."

*P12L1: Why does table 5 show the median whereas otherwise only the ensemble mean is discussed or shown. (Using only the median would be perhaps a better option overall)*

We present mean values in other estimates because we also discuss the standard deviation to the mean. Medians are shown only in this table (Table 8 in the revised manuscript), in order to provide more robust estimates of the model error and sampling bias for each region. The following sentence has been added to clarify this point:

"The sampling bias is evaluated using the median of the multiple models to provide robust estimates of the model performance."

*P13L8: The ozone network in SH high-latitudes is actually quite high because of the need to monitor the ozone hole. The launch frequency varies for some stations a lot because more sondes are launched during the ozone hole season.*

Yes, I agree. However, because only three stations were considered in the comparison following Tilmes et al. (2012), the ozonesonde network is not sufficient to capture the ozone variations. Our results suggest that the temporal sampling bias mainly causes the sampling bias in the SH high latitudes in MAM and JJA. This is discussed in the revised manuscript.

*P13L24: Please add also the sampling biases for the latitude bands.*

Added in Table 10. The following sentences have been added to discuss the results:
"Because the seasonal variations differ among different regions, the seasonal amplitude estimated for the entire NH extratropics (30-90N) is largely different between the two estimates throughout the troposphere."
"The sampling bias in the seasonal amplitude estimated for the entire tropics is larger than 60 % throughout the troposphere both in the NH (Eq-30N) and SH (30S-Eq)."

*P14 L14: Please clarify how much of the analysis ensemble spread depends on some-times arbitrary choices to cause spread between the ensemble members.*

The following sentence has been added:
"Note that the data assimilation setting influences the analysis uncertainty estimation in the reanalysis. In particular, the analysis spread was found to be sensitive to the choice of ensemble size (Miyazaki et al., 2012b). A large ensemble size is essential to capture the proper background error covariance structure (i.e., analysis uncertainty)."

*P14L15: Please clarify to what extent the analysis uncertainty is controlled by the uncertainty of the assimilated observations.*

The sentences have been rewritten as follows:
"Miyazaki et al (2015) investigated that the analysis spread is caused by errors in the model input data, model processes, and assimilated measurements, and it is reduced if the analysis converges to a true state. The analysis spread is smaller in the extratropical lower stratosphere than in the tropical upper

troposphere at 200 hPa, because of the high accuracy of the MLS measurements. In contrast, in the middle troposphere, the analysis spread is generally smaller in the tropics than the extratropics because of the higher sensitivities in the TES retrievals."

*P14L21: How does the ensemble spread relate to the spread of the ensemble in the EnKF. Could the ACCMIP ensemble spread be used to verify the EnKF ensemble spread?*

The simultaneous enhancement of the analysis uncertainty and the model spread indicates low robustness of the validation results, as discussed in the manuscript. Verification of EnKF ensemble spread using ACCMIP ensemble model spread would be an interesting research topic but requires careful discussion and is clearly out of the scope of the present study.

*P15L3: Please see my general comment on this chapter. Improvement on species not directly assimilated needs to be demonstrated. Long-term reanalysis could suffer from artificial jumps because of the change in the observing system (for example degradation of TES after 2010).*

Corrections have been made. Please see our reply above.

*P15L29: I don't understand this conclusion at all. Re-analysis are only valid for present day conditions when observations are available. They cannot be used to for pre- industrial estimates nor the differences with today's values.*

Assuming that the persistent systematic bias from the pre-industrial to present day can be attributed to time-independent model errors in chemical and transport processes, as suggested by previous studies (e.g., Young et al., 2013), the validation results using the reanalysis for the present day has the potential to evaluate preindustrial to present-day ozone radiative forcing. This is discussed in the manuscript.

*P16L4: Please mention that you (only) consider ozone sondes as reference in this paper.*

The following sentence has been added:
"The evaluation results are also used to quantify the ozonesonde network sampling bias."

*P16L6: Please mention the advantage of using a re-analysis, i.e. a gridded field. Please mention that the biases of the re-analysis against ozone sondes are small.*

The sentences have rewritten as follows:

"The reanalysis product provides comprehensive and unique information on global ozone distributions for the entire troposphere and on the weakness of the individual models and multi-model mean. Validation of the chemical reanalysis using global ozonesondes shows good agreement throughout the free troposphere and lower stratosphere for both seasonal and year-to-year variations."

*P16L17: Please add a statement if these finding are consistent with other evaluation studies, i.e. the Young et al. paper.*

The following sentence has been added:

"The performance of the ACCMIP model when compared with the reanalysis is qualitatively similar for most cases from that shown by Young et al. (2013) using the ozonesonde measurements but quantitatively different because of the ozonesonde network sampling bias."

*P16L20-30: Please give some numbers for the sampling biases. Also include the sampling bias w.r.t to latitude bands.*

The following sentences have been added:

"For the global tropics, the ozonesonde sampling bias is largely negative by 80 % in the NH (Eq-30N) in SON and by 50 % in the SH (30S-Eq) in MAM."

"Large sampling biases (> 60 %) exist in the SH extratropics (90-30S) in DJF and MAM"

*p17L4: Please add a statement that it will be a challenge to combine all these observations in a consistent way in a more long term re-analysis.*

The following sentence has been added:

"Combining many observations requires a bias correction procedure for each assimilated measurement to improve the reanalysis quality but needs to be carefully checked."

---

## Author Comment (AC2) · 5 Apr 2017

**Authors' comments in reply to the anonymous referee for "Evaluation of ACCMIP ozone simulations using a multi-constituent chemical reanalysis" by K. Miyazaki and K. Bowman**

We want to thank the referee for the helpful comments. We have revised the manuscript according to the comments, and hope that the revised version is now suitable for publication. Below are the referee comments in italics, with our replies in normal font.

*Reply to Referee #2*

*Specific Comments:*

*Abstract: Please mention that a 4-year reanalysis data is used to compare ACCMIP time-slice simulations, and that the evaluation itself can be biased because of this inconsistency.*

The abstract has been rewritten to describe the time period of the reanalysis and ACCMIP simulations.

*P1L17: Insert anthropogenic between "...important greenhouse.."*

Inserted.

*P2L1: There are several studies highlighting the use of CTMs/CCMs to assess the radiative impacts of tropospheric ozone prior to Bowman et al (2013). Please acknowledge those.*

Several studies are cited in the following sentence.

*P2: 2nd and 3rd paragraphs discuss uncertainties in measurements for evaluating chemistry-climate models. I think they can be combined and modified for clarity.*

Combined and modified.

*P3L12: Please provide motivation for evaluating only tropospheric ozone and not its precursors (e.g., NO2, CO). Presumably biases in ozone are driven by biases in its precursors.*

The following sentences have been added:
"Model errors in precursors can also be evaluated using the reanalysis product, and this could help

identify error sources in tropospheric ozone simulations. However, because no other study has shown the potential of reanalysis ozone for model evaluation, this study focuses on tropospheric ozone only."

*P4. How is photolysis calculated in the forecast model? Does the model represent methane - prescribed concentrations or emissions? Since the ACCMIP models used different emissions inventory for ozone precursors compared to what is used in this reanalysis product, how would the comparison be affected by this difference?*

The following sentences have been added:

"The radiative transfer scheme considers absorption within 37 bands, scattering by gases, aerosols, and clouds, and the effect of surface albedo. Detailed radiation calculations are used for photolysis calculation. Methane concentrations were scaled on the basis of present-day values with reference to the surface concentration."

Because the surface emissions of NOx and CO are optimized using data assimilation in the chemical reanalysis, the difference in the emission inventory should not affect the comparison. In our previous studies (Miyazaki et al., 2012, 2015, 2017), it was confirmed that the a priori emissions do not largely influence the a posteriori emissions through long data assimilation cycles.

*P4L13: What is the convection scheme of MIROC-AGCM? Please describe it in a sentence.*

The sentence has been rewritten as follows:

"Lightning NOx (LNOx) sources in MIROC-Chem were calculated based on the relationship between lightning activity and cloud top height (Price and Rind, 1992) and using the convection scheme of MIROC-AGCM developed based on the scheme presented by Arakawa and Schubert (1974)."

*P5L7: What is DOMINO data? What does the acronym stand for?*

DOMINO stands for Dutch OMI NO2. This is described in the revised manuscript.

*P5L21: According to Lamarque et al (2013), models were run for several years (up to 10 years) for each time slice. How are the model results for 2000 time-slice handled for comparison with the reanalysis?*

The following sentence has been added:

"The number of years that the ACCMIP models simulated for the 2000 decadal simulation mostly varied between 4 and 12 years for each model. Each model simulation was averaged over the simulated years."

To discuss the limitation of the short period of the chemical reanalysis used for the validation, the following sentences have been added in Section 6.1:

"The five-year reanalysis (2005-2009) may cause biases in the estimated model errors in the evaluation of the 2000 decade ACCMIP simulations that used decadal-averaged SST boundary conditions and biomass burning emissions averaged over 1997-2006 (Lamarque et al., 2010). It may neglect the influences of interannual and decadal changes in anthropogenic and biomass emissions and meteorology. Longer-term reanalysis and time-consistent validation are required to obtain more robust error estimations."

*P5L29: Young et al. (2013) note that although the models used the same anthropogenic and biomass burning emissions, model-to-model diversity in the implemented chemical scheme resulted in differences in precursor (especially VOCs) emissions across the models. So, the statement "same emissions were used in all the models" is not accurate. Please modify.*

Removed.

*P6L7: Please provide details on how the ACCMIP model monthly ozone concentrations were interpolated to 2 hour temporal resolution? What diagnostics from the reanalysis were used to compare with observations - monthly or hourly ozone? Some clarification is needed here.*

The sentences have been rewritten as:

"The two-hourly reanalysis and forecast model (i.e., control run) fields were linearly interpolated to the time and location of each measurement, with a bin of 25 hPa, and then compared with the measurements. For the ACCMIP models, the monthly model outputs were compared with the measurements at the location of each measurement."

*P6L29: Can you please elaborate on the model setting that caused this significant degradation of the representation of ozone in the UTLS?*

The clause has been rewritten as follows:

"because of different model settings, such as the upper boundary conditions of NOy, Cly, and Bry"

*P7L3: The terminology to refer to model output is somewhat confusing making it difficult to keep track of observations versus model output. Suggest referring to output from the forecast model and ACCMIP as "forecast model output" and "ACCMIP model output", respectively, and observations as "data".*

Corrected throughout the paper.

*P7L22-24: The simulation of the "wave-1" pattern by ACCMIP models has already been highlighted by Young et al. (2013). This reference needs to be cited.*

Cited.

*P7L32: It is not clear what "common reported" means here. Please clarify.*

Replaced by "Young et al. (2013) consistently revealed the positive bias in the NH and negative bias in the SH using OMI/MLS tropospheric ozone column measurements."

*P8L10: Which region is "In this region. . ." referring to?*

Replaced by "In the NH extratropics in the lower and middle troposphere".

*P8L14-15: These results are consistent with those discussed in Young et al. (2013).*

Young et al. (2013) is cited.

*P8L24: The differences could also be associated with the way biomass burning emissions are handled across models - whether they are emitted at the surface layer or distributed vertically in the model, as a larger role for biomass burning in the tropical mid to upper troposphere has been suggested recently (Anderson et al., 2016).*

The following sentence has been added:
"For instance, biomass burning emissions are handled differently across the models, which may lead to differences in ozone simulations in the tropics (Anderson et al., 2016)."

*P8L33-34: Why are results from a specific model highlighted? Is it because model 8 is driving the large model diversity at 200hPa? Please provide a figure to support this statement if this is indeed the case.*

The sentence has been removed.

*P9L3: It took me a while to understand the meaning of "reanalysis concentrations from the ozonesonde sampling". Please rephrase this to indicate that reanalysis output is sampled at ozonesonde sites instead of averaging the reanalysis at all grid cells.*

Replaced.

*P929-31: Here again the results are consistent with Young et al. (2013) - see their Figure 4 and its discussion.*

Cited.

*P10L23-25: Please clarify what is meant by "radiative heating distribution in chemistry-climate simulations are largely uncertain." Are you referring to the radiative heating due to tropospheric ozone? Please elaborate on how the O3 NH/SH ratio provides information on the radiative heating distribution.*

The sentence has been replaced by
"The large systematic error in the NH/SH ratio suggests that, for instance, the inter-hemispheric distribution of radiative heating due to tropospheric ozone in chemistry--climate simulations are largely uncertain in most models"

*P11L17: replace "completion" with compilation.*

Replaced.

*P11L19: By coarse resolution, do you mean the coarse horizontal resolution?*

Yes. Replaced.

*P11L29: Did you mean - "...less variabilities in the SH than in the NH"?*

Yes. Corrected.

*P12L9-11: Please quantify large in "Large negative model biases..." and larger in "...errors are larger than those. . ."*

The sentences have been rewritten as:

"Large negative model biases against the ozonesonde observations have been reported by Young et al. (2013) for 250 hPa (by about -13 % for the NH polar west and -18 % for the NH polar east for the annual mean concentration), whereas results from this study suggest that these errors based on the ozonesonde sampling (by -14 % for the NH polar west and -18 % for the NH polar east in DJF in our estimates) are larger than those from regional and seasonally representative model bias (by -3 % and +5 %, respectively)."

*Section 5.1 Please elaborate on how the comparisons discussed here may be influenced by inconsistency in the time period of the reanalysis and the ACCMIP simulations. The reanalysis output is for the 2005-2009 whereas the ACCMIP simulations are representative of 2000. Additionally, the precursor emissions used in ACCMIP simulations were decadal means and not specific to the year of simulation. For example, year 2000 biomass burning emissions were calculated as average over 1997–2006 (Lamarque et al., 2010), so they encompass the high emissions over Southeast Asia in 1998, an El Nino year. One would expect that biases due to sampling in time would occur similar to the biases due to sampling in space. How significant are the biases due to spatial sampling errors (highlighted here) compared to temporal sampling errors. The issue of temporal and spatial sampling was recently highlighted by Lin et al. (2016) in the context of tropospheric ozone trends.*

The limitation of the evaluation based on the 2005-2009 reanalysis fields is discussed in the revised manuscript. Please see our reply above.

The spatial and temporal sampling errors are separately evaluated in the revised paper, and the results are presented in Table 7 and discussed in Section 5.1. Lin et al. (2016) is cited in the revised paper.

*P14L24: Replace biogenetic with biogenic.*

Replaced.

*P15L6: I am not sure if the 2005-2009 can be considered a "long-record" Are the authors referring to*

*the possibility of a long-record reanalysis sometime in the future when observations have accumulated in time.*

In this sentence, we are referring to the possibility in the future. The sentence has been replaced by "A long-record of the reanalysis will allow...."

*P15L7-9: From Lamarque et al. (2013): "This averaging was designed to reduce the effect of interannual variability and therefore provide optimal conditions from which average composition changes and associated forcings can be more readily computed." The ACCMIP simulations were designed to remove interannual variability, therefore, it is unjustified to state that "the influence of ENSO was not well-simulated in ACCMIP..."*

The sentence has been replaced by
"It is noted that the influence of ENSO was not included..."

*P33 Figure 5. Please provide statistics such as mean bias and correlation for the comparisons here.*

These statistics are provided in Table 4.

---

## Author Response (AR2)

**Authors' comments in reply to the anonymous referee for "Evaluation of ACCMIP ozone simulations and ozonesonde sampling biases using a satellite-based multi-constituent chemical reanalysis" by K. Miyazaki and K. Bowman**

We want to thank the referee for the helpful comments. We have revised the manuscript according to the comments, and hope that the revised version is now suitable for publication. Below are the referee comments in italics, with our replies in normal font.

***Reply to Referee #1***

*Abstract:*

*P1 L18: define better the relative sampling bias, i.e relative to what?)*

The sentences have been rewritten as follows:

"To estimate ozonesonde sampling biases, we computed model bias separately for global coverage and the ozonesonde network. The ozonesonde sampling bias in the evaluated model bias for the seasonal mean concentration relative to global coverage…"

*P1L19: Also give information about typical sampling biases in other regions.*

The following sentences have been added:

"…, and up to 80 % for the global tropics. In contrast, the ozonesonde sampling bias is typically smaller than 30 % for the Arctic regions in the lower and middle troposphere."

According to these changes, the abstract has been reformulated.

*P1 L8 better "emissions sources"*

Replaced.

*P1 L25 – add . after al*

Added.

*P1 L28 "data do"*

Corrected.

*P3 L2 add "strong" between not and enough*

Added.

*P3 L16 f. Inness et al. 2013 use a coupled system IFS-MOZART. Flemming et al. 2017 use C-IFS with CB05 chemistry in their re-analysis.*

Corrected.

*P5 l31, at . after al*

Added.

*P11L25 Define "seasonal amplitude"*

Already defined in P11L9.

*P17L33. Please discuss Fig. 11 in more detail, especially the analysis errors and their relation to the ACCMIP ensemble spread. Why is the ensemble spread at 200 hPa so much larger than the re-analysis error? How do the different tropopause definitions impact the ozone sampling error?*

The following sentences have been added to discuss the relationship between the analysis uncertainty and model spread:

"Meanwhile, the magnitudes of the model spread and analysis uncertainty differ considerably for some regions. At 200 hPa and higher in the extratropics, the analysis uncertainty is smaller than the model spread, where the reanalysis fields are strongly constrained by MLS measurements. These results suggest that further improvements, for instance, on the representation of the tropopause and STE are required for some of the models, in order for the reanalysis and ensemble models to have similar levels of uncertainty. In the lower troposphere, in contrast, the larger analysis uncertainty than the ensemble spread suggests that further observational constraints are required for the reanalysis for a fair comparison."

The different representations of the tropopause and STE could affect model errors and thus the evaluated sampling bias, which have already been discussed in the manuscript.

*P19 L33: define sampling biases. Say what you did to get to relative numbers. Also summarise the sampling biases for the regions with low to moderate sampling biases (see Figure 9).*

The sentences have been rewritten as follows:
"To estimate the sampling biases in the ACCMIP model evaluation, we compared two evaluation results of the mean model bias, using the chemical reanalysis based on the complete and ozonesonde samplings. For instance, the ozonesonde sampling bias in the evaluated model bias (relative to the model bias for the complete sampling) is…"

The following sentence has been added to summarize low sampling biases:
"The ozonesonde sampling bias is typically smaller than 30 % for the NH polar regions except in boreal winter and over the equatorial Americas, the Atlantic Ocean and Africa, and at the SH mid-latitudes in austral winter and spring from the lower to middle troposphere."

*Table 2*
*Should be Legionowo not Legionowa*
*Spell out Hohenpbg*

Corrected.

*Figure 1:*
*Mention in caption that you show relative errors, also relative to what*

Mentioned.

*Figure 5:*
*Clarify if the models are shown at "complete" or "ozone-sonde" sampling*

The following sentence has been added:
"The ACCMIP model results are shown for the average over all model grid points."

*Figure 6:*

*Please add also the seasonal amplitude from the ozone-sonde sampling as this is also done in the other figures (e.g. Figure 7). Explain averaging interval for the seasonal amplitude (i.e. monthly values)*

The seasonal amplitude from the ozonesonde sampling has been added. The time interval is explained in the revised manuscript.

[revised manuscript text omitted]